# Franckeite as a naturally occurring van der Waals heterostructure

Aday J. Molina-Mendoza[1,*], Emerson Giovanelli[2,*], Wendel S. Paz[1], Miguel Angel Niño[2], Joshua O. Island[3,†], Charalambos Evangeli[1,†], Lucía Aballe[4], Michael Foerster[4], Herre S.J. van der Zant[3], Gabino Rubio-Bollinger[1,5], Nicolás Agraït[1,2,5], J.J. Palacios[1], Emilio M. Pérez[2] & Andres Castellanos-Gomez[2]

The fabrication of van der Waals heterostructures, artificial materials assembled by individual stacking of 2D layers, is among the most promising directions in 2D materials research. Until now, the most widespread approach to stack 2D layers relies on deterministic placement methods, which are cumbersome and tend to suffer from poor control over the lattice orientations and the presence of unwanted interlayer adsorbates. Here, we present a different approach to fabricate ultrathin heterostructures by exfoliation of bulk franckeite which is a naturally occurring and air stable van der Waals heterostructure (composed of alternating $SnS_2$-like and PbS-like layers stacked on top of each other). Presenting both an attractive narrow bandgap ($<0.7\,eV$) and p-type doping, we find that the material can be exfoliated both mechanically and chemically down to few-layer thicknesses. We present extensive theoretical and experimental characterizations of the material's electronic properties and crystal structure, and explore applications for near-infrared photodetectors.

[1] Departamento de Física de la Materia Condensada, Universidad Autónoma de Madrid, Campus de Cantoblanco, E-28049 Madrid, Spain. [2] Instituto Madrileño de Estudios Avanzados en Nanociencia (IMDEA-Nanociencia), Campus de Cantoblanco, E-28049 Madrid, Spain. [3] Kavli Institute of Nanoscience, Delft University of Technology, Lorentzweg 1, 2628 CJ Delft, The Netherlands. [4] ALBA Synchrotron Light Facility, Carrer de la Llum 2-26, Cerdanyola del Vallés, Barcelona 08290, Spain. [5] Condensed Matter Physics Center (IFIMAC), Universidad Autónoma de Madrid, E-28049 Madrid, Spain. * These authors contributed equally to this work. † Present addresses: Department of Physics, University of California, Santa Barbara, California 93106, USA (J.O.I.); Department of Physics, Lancaster University, Lancaster LA1 4YB, UK (C.E.). Correspondence and requests for materials should be addressed to J.J.P. (email: juanjose.palacios@uam.es) or to E.M.P. (email: emilio.perez@imdea.org) or to A.C.-G. (email: andres.castellanos@imdea.org).

The demonstration of the deterministic placement of 2D crystals has opened the door to fabricate more complex devices[1], but more importantly, it has started the investigation to tailor the properties of designer materials by stacking different 2D crystals to form the so-called van der Waals heterostructures[2]. One approach to produce such heterostructures is to use epitaxially grown materials assembled sheet by sheet[3]. This method, however, remains challenging and has only been demonstrated for a few van der Waals heterostructures so far[4–6]. Another approach is the manual assembly of individual layers obtained by mechanical exfoliation from bulk and the deterministic placement of one layer on top of another[7–12]. This method also presents issues that remain to be solved, such as controlling the exact crystalline alignment between the stacked lattices and avoiding the presence of interlayer atmospheric adsorbates.

Here, we present the study of ultrathin layers of franckeite, a naturally occurring sulfosalt with a structure formed by alternated stacking of tin disulfide-based ($SnS_2$) and lead sulfide-based (PbS) layers. Interestingly, the individual layers present a larger bandgap than the naturally formed van der Waals heterostructure ($< 0.7$ eV), which is among the narrowest found in 2D semiconductors. We also find that franckeite is a p-type material, a very rare feature in two-dimensional semiconductors, only found so far in a few materials such as black phosphorus and tungsten diselenide[13–17]. But unlike black phosphorus, franckeite is air-stable. We combine density functional theory (DFT) calculations with the experimental characterization of the optical and electrical properties of ultrathin franckeite, which we isolate both by micromechanical cleavage and liquid-phase exfoliation, to offer a complete picture of its unique properties.

## Results

**Franckeite crystal and band structure.** Franckeite is a layered material from the sulfosalt family formed by the stacking of pseudohexagonal (H) and pseudotetragonal (Q) layers that interact by van der Waals forces[18–20]. The Q layer is composed of four atomic layers of sulfide compounds with the formula MX, where $M = Pb^{2+}$, $Sn^{2+}$ or $Sb^{3+}$ and $X = S$. The H layer consists of octahedrons of disulfide compounds with the formula $MX_2$, where $M = Sn^{4+}$ or $Fe^{2+}$ and $X = S$. In Fig. 1a, we show the

crystal structure of the material, indicating the different atomic layers present in the crystal.

In order to gain insight into the expected material electronic properties, we first compute the band structure based on the crystal structure described above. To this aim, we perform DFT calculations as implemented in the QuantumEspresso code[21]. The franckeite structure exhibits a long-range one-dimensional transversal wave-like modulation and a non-commensurate layer match in two dimensions[18]. This will not be taken into account since it would be computationally too expensive and it is expected to only introduce minor corrections to the results.

We first investigate the band structure of individual Q and H layers (Fig. 1b and c, respectively). In these calculations we only consider PbS and $SnS_2$ compounds, ignoring the influence of substitutional Sb or Fe atoms. For the H layer (Fig. 1b), using the standard generalized gradient approximation (GGA) to the functional, we obtain an indirect bandgap at X of $\sim 1$ eV that increases to 2 eV using HSE06. For the Q layer (Fig. 1c), our calculations yield a semimetal with a gap at X of $\sim 0.5$ eV using GGA and $\sim 0.7$ eV with HSE06. The band structure of the bulk crystal (the thickness dependent band structure is discussed in Supplementary Note 1 and Supplementary Fig. 1) formed by alternate stacking of Q and H layers with the same composition as that of the individual layers, results in two sets of bands separated by a small gap at X of $\sim 0.5$ eV (GGA, Fig. 1d), while with HSE06 we obtain a gap of $\sim 0.75$ eV (we direct the reader to Supplementary Notes 2 and 3, Supplementary Table 1 and Supplementary Fig. 2 for more details about the calculations of the bandgap). Although the Fermi level lies below the valence band, and we cannot properly speak about the existence of a gap, a further investigation of the role played by substitutional Sb atoms shows that the Fermi level is shifted upwards when the material is doped with Sb (we direct the reader to the Supplementary Figs 3–5 and Supplementary Note 4 for calculations including Sb substitutional atoms). These results can be compared with the electronic bandgap measured by means of scanning tunneling spectroscopy (STS) on bulk franckeite crystals, where we observe an electronic bandgap of $\sim 0.7$ eV with the valence band edge closer to the Fermi level, further confirming the p-type doping of the material (also confirmed by the thermopower measurements). We direct the reader to the Supplementary Notes 5 and 6 and Supplementary Figs 6 and 7 for

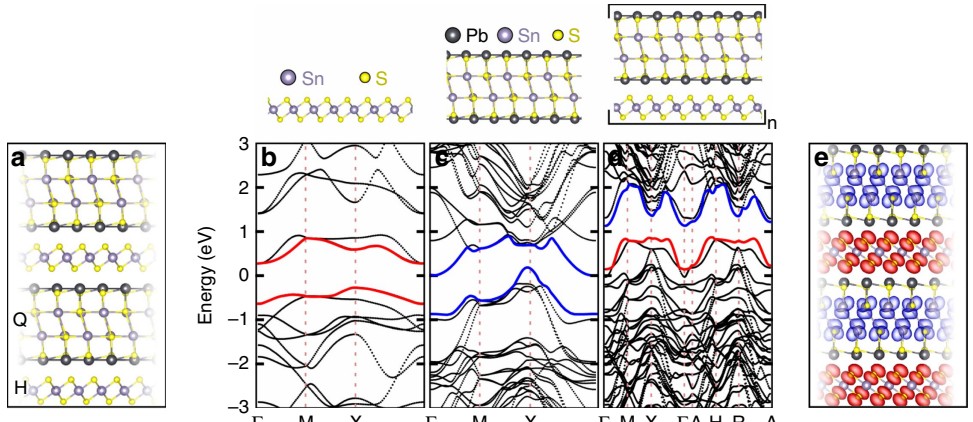

**Figure 1 | Franckeite crystal and band structure.** (**a**) Crystal structure of franckeite: the Q layer includes MX compounds, where $M = Pb^{2+}$ or $Sn^{2+}$ (M can also be $Sb^{3+}$ replacing $Sn^{2+}$) and $X = S$, while the H layer includes $MX_2$ compounds, where $M = Sn^{4+}$ (M can also be $Fe^{2+}$ replacing $Sn^{4+}$) and $X = S$. (**b**) GGA band structure of the H layer that presents a bandgap of $\sim 1$ eV. (**c**) Band structure of the Q layer which presents a bandgap of $\sim 0.5$ eV. (**d**) Band structure of the franckeite crystal that presents a bandgap of $\sim 0.5$ eV. The valence band states are provided by the H layer (red line), while the conduction band is given by the Q layer (blue line), suggesting that franckeite is a type-II heterostructure. (**e**) Bloch states in franckeite in which the valence (red) and conduction (blue) bands are represented.

more details on the STS and thermopower measurements, as well as X-ray diffraction characterization.

In the band structure shown in Fig. 1d, we note that the valence band wavefunctions correspond to the H layer while the conduction band wavefunctions belong to the Q layer, much in analogy with an artificial type-II semiconducting heterostructure[22]. In Fig. 1e we show a projection of the crystal structure with the corresponding Bloch states, where blue areas represent the Bloch states corresponding to the blue band in Fig. 1d and the red areas represent the Bloch states in the red band in Fig. 1d.

**TEM and XPS of mechanically exfoliates flakes**. Transmission electron microscopy (TEM) of mechanically exfoliated franckeite

flakes reveal both the high degree of orientation in the stacking of the H and Q layers and its misfit structure. Figure 2a shows a low magnification TEM image of a flake with regions of different thicknesses, as inferred from the difference in TEM contrast. The TEM image shows regularly spaced fringes which are due to the corrugation of the crystal induced by the interaction between the misfit Q and H layers, as demonstrated by Makovicky et al.[18] with cross-sectional TEM in bulk franckeite. Figure 2b shows a high resolution (HRTEM) image where the atoms of both the Q and H layers can be resolved. The corresponding selected area electron diffraction (SAED) diagram (Fig. 2c) has consequently been indexed considering franckeite as a stacking of the Q and H layers, respectively described as tetragonal and orthohexagonal[23]. We address the reader to Supplementary Figs 8 and 9 for

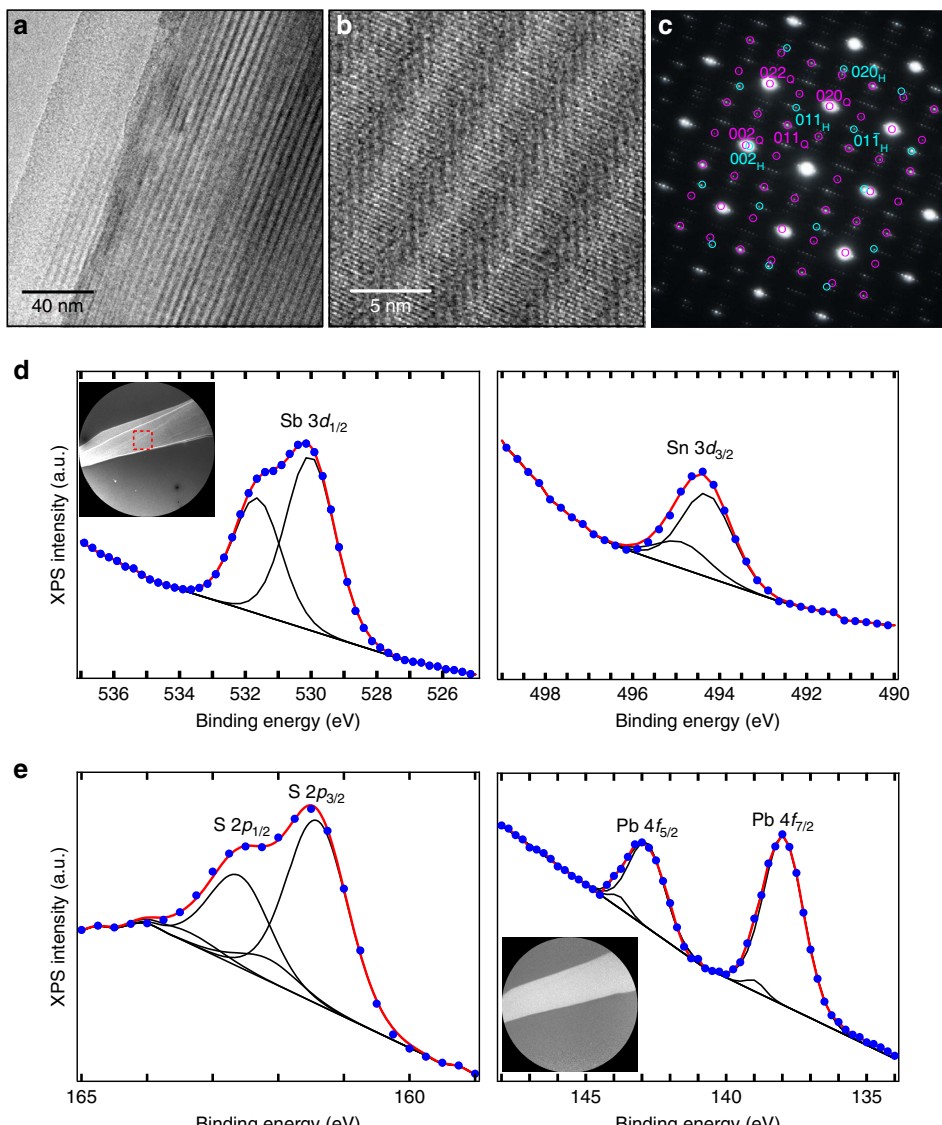

**Figure 2 | Characterization of mechanically exfoliated franckeite flakes.** (**a**) HRTEM micrograph of a franckeite sheet exhibiting the characteristic fringes of franckeite due to the corrugation induced by the misfit between Q and H layers. The scale bar is 40 nm. (**b**) Representative atomic scale HRTEM of an ultrathin franckeite layer. The scale bar is 5 nm. (**c**) SAED diagram consistent with a misfit layer compound made of PbS and $SnS_2$ layers: Q (purple) and H sublattices (light blue) lead to the most intense reflections on which superlattice rows of weak intensity are centred. The diagram has been indexed using tetragonal and orthohexagonal vectors for the Q and H phases respectively, according to the orientation and nomenclature defined in ref. 23. (**d**) Sb $3d_{3/2}$ and Sn $3d_{3/2}$ XPS spectrum acquired with photon energy $h\nu = 600$ eV. Inset: LEEM image (the field of view is 50 μm and the electron energy is 0.12 eV), the red square indicates the region of integration where the XPS spectra has been acquired. (**e**) S $2p_{1/2}$ and $2p_{3/2}$ and Pb $4f_{5/2}$ and $4f_{7/2}$ XPS spectrum acquired with photon energy $h\nu = 230$ eV. Inset: XPEEM image at Pb $4f_{7/2}$ core level (the field of view is 50 μm and the photon energy is 230 eV). The strong background in the XPS spectra is due to the tail of secondary electrons cascade.

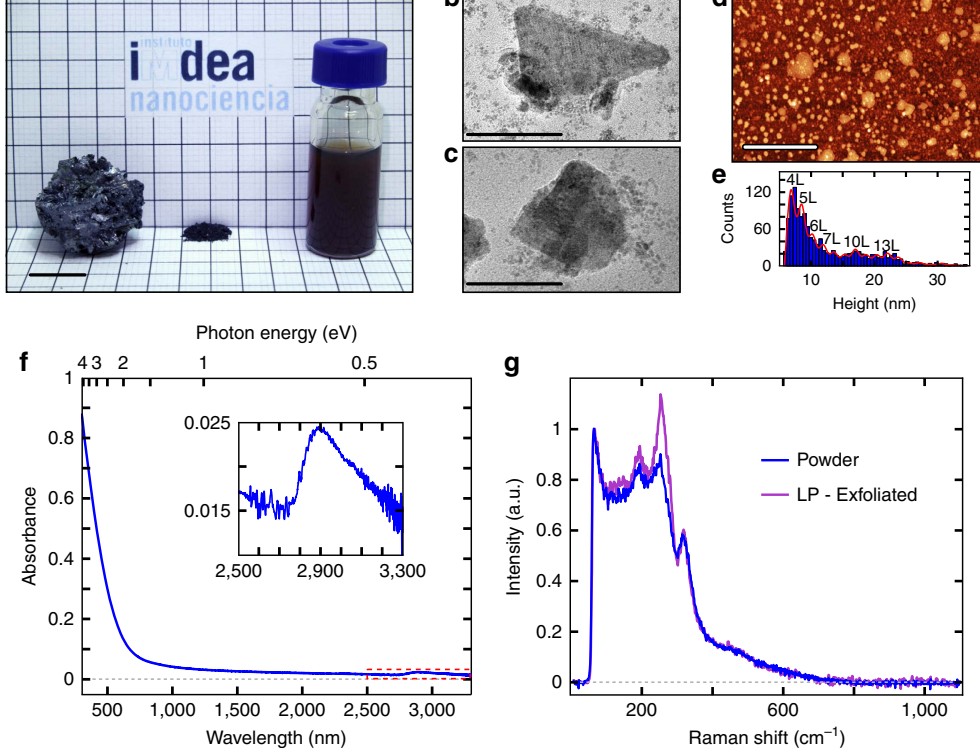

**Figure 3 | Liquid phase exfoliation of franckeite.** (**a**) Franckeite samples. Left: bulk mineral; middle: powder material obtained after grinding of raw chips; right: suspension of exfoliated material prepared by sonication of a $100\,mg\cdot ml^{-1}$ powder dispersion in NMP. The scale bar is 1 cm. (**b,c**) TEM images of representative franckeite nanosheets prepared by exfoliation of a $100\,mg\cdot ml^{-1}$ powder dispersion in NMP. The scale bars are 150 and 75 nm, respectively. (**d**) AFM topographic characterization of franckeite nanosheets obtained from the exfoliation of a $1\,mg\cdot ml^{-1}$ powder dispersion in isopropanol/water 1/4 (v/v). The scale bar is 2 μm. (**e**) Statistical analysis of the AFM raw height data. The inserted numbers indicate the corresponding number of layers (unit cell, H + Q layer, 1.7 nm in thickness) from ~4 layers (4 L) up to ~13 layers (13 L). (**f**) UV-Vis-NIR spectrum of a thin film of franckeite colloidal suspension deposited on a glass slide; the sample originates from the liquid-phase exfoliation of a $100\,mg\cdot ml^{-1}$ franckeite powder dispersion in NMP. Inset: zoom of the region indicated by a dashed red line that highlights the absorption peak around 2,900 nm. (**g**) Raman spectra of franckeite raw powder (blue line) and liquid-phase (LP)-exfoliated franckeite obtained from the sonication of a $100\,mg\cdot ml^{-1}$ powder dispersion in NMP (pink line).

scanning electron and atomic force microscopy images of mechanically exfoliated flakes.

We have also characterized mechanically exfoliated franckeite flakes using synchrotron micro-X-ray photoemission spectroscopy (XPS) with lateral resolution of ~20 nm in a photoemission electron microscope (PEEM) (ref. 24). The flakes were transferred onto a metallic Pt surface to avoid sample charging. XPS-PEEM spectra of the four main components of franckeite Sb 3$d$, Sn 3$d$, S 2$p$ and Pb 4$f$ are displayed in Fig. 2d and e. From the core level fits we distinguish two components for each element: for Sn, we assign the two components to $Sn^{2+}$ and $Sn^{4+}$ from the Q and H structural layers of franckeite, respectively. The Sb 3$d$ core level also presents two components: $Sb^{3+}$ appears in the Q layer with an extra Sb in a different environment (see more details in the Supplementary Note 7 and Supplementary Figs 10–12, as well as bulk franckeite XPS spectra). The ratio between Sb and Sn intensities, after normalization to the photoemission cross section and to the microscope transmission, results in an excess of 33% of Sb over Sn. The S 2$p$ core level also indicates two different environments, PbS and $SnS_2$, while Pb 4$f$ has a strong doublet (PbS) and another minor component, possibly due to some lead oxide. In the inset of Fig. 2d we show a low energy electron microscopy (LEEM) image of the flake, appearing as a bright stripe crossing the field of view, with some steps running nearly parallel to one of its edges. From the XPS-PEEM chemical images (inset of Fig. 2e, measured at the Pb 4$f_{7/2}$ core level peak), we conclude that franckeite is chemically homogeneous: the image

shows a uniform bright stripe through the whole flake, with the same behaviour for all the other chemical constituents.

**Liquid-phase exfoliation.** Liquid-phase exfoliation (LPE) of layered materials allows scale-up, improves processability, and opens the way to chemically functionalize the nanosheets in suspension[25]. We demonstrate LPE isolation of franckeite by bath ultrasonication for 1 h at 20 °C, using franckeite powder obtained from careful grinding of mineral pieces (Fig. 3a). First, LPE was carried out in $N$-methyl-2-pyrrolidone (NMP), already used to exfoliate graphite and transition metal dichalcogenides[26,27] due to its surface tension and coordination properties, which favour layer separation and subsequent nanosheet stabilization. LPE was also investigated in various isopropanol/water (IPA/water) mixtures, as well as in two other polar solvents, methanol and $N,N$-dimethylformamide (DMF). In the following, we will focus on NMP and IPA/water 1/4 (v/v), the latter matching surface tension with $SnS_2$ (ref. 28). Details on LPE in the other solvents and their comparison are reported in Supplementary Note 8.

Remarkably, all the suspensions prepared in NMP were indefinitely stable in time (>6 months, Supplementary Fig. 13). Analysis of the most concentrated colloids using atomic force microscopy (AFM, Supplementary Fig. 14) and TEM (Fig. 3b,c, and Supplementary Fig. 15) reveals large flakes (~100 nm in lateral size) of height <25 nm, together with copious amounts of very small fragments (~10 nm–20 nm in lateral size), less present

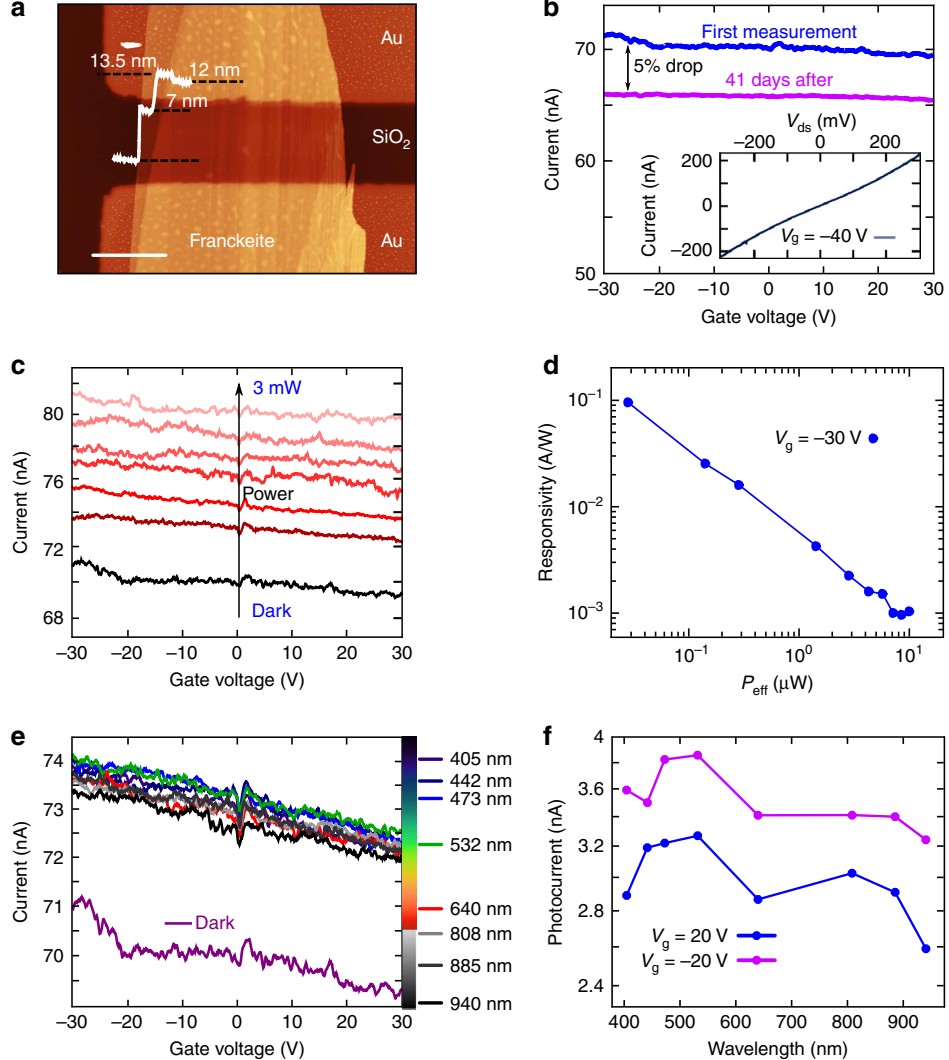

**Figure 4 | Franckeite-based nanodevices. (a)** AFM topographic image of a franckeite flake deposited on $SiO_2$ substrate with pre-patterned Ti/Au electrodes. The thickness of the flake ranges from 7 nm ($\sim$4 layers) to 13.5 nm ($\sim$8 layers). The scale bar is 7 µm. **(b)** Current as a function of the applied back-gate voltage in dark conditions for the device shown in **a** ($V_{ds} = 150$ mV). The gate-dependence shows a p-type doping, hole conduction. The first measurement (blue line) was repeated after 41 days (pink line), showing a drop of 5%, yielding a good stability of the device. Inset: current-voltage curve with an applied back-gate voltage of $-40$ V. **(c)** Current as a function of the applied back-gate voltage ($V_{ds} = 150$ mV) for the device shown in **a** in dark conditions and upon illumination with a 640 nm wavelength laser with different powers. **(d)** Responsivity of the device shown in **a** upon illumination with a 640 nm wavelength laser as a function of the laser effective power with an applied back-gate voltage of $V_g = -30$ V and $V_{ds} = 150$ mV. **(e)**, Current as a function of the applied back-gate voltage ($V_{ds} = 150$ mV) for the device shown in **a** upon illumination with lasers of different wavelengths at the same intensity ($P_d = 6.3$ mW·cm$^{-2}$). There is photocurrent generation even at wavelengths as large as 940 nm. **(f)** Photocurrent as a function of the laser wavelengths with the same light intensity for back gate voltages of $-20$ V and $+20$ V and $V_{ds} = 150$ mV.

when exfoliating a less concentrated dispersion (Supplementary Fig. 16).

LPE in IPA/water leads to much more uniform nanosheets (see Supplementary Figs 17–25 for the complete series). As an example, Fig. 3d shows a large area AFM topographic image obtained after drop-casting and drying of a franckeite suspension in IPA/water 1/4 on a mica substrate. Statistical analysis of the height data (Fig. 3e) attests to the formation of very thin flakes, with a narrow thickness distribution between 6 and 12 nm. This range corresponds to a maximum number of $\sim$4–7 franckeite layers (being one layer an H-Q pair), without considering adsorbed solvent molecules that might increase the measured thickness by up to 1.2–1.3 nm (ref. 29). The corresponding lateral size distribution (Supplementary Fig. 23) is also more homogeneous and displaced towards larger sheets (*c.a.* 200 nm).

LPE in methanol or DMF (Supplementary Figs 26–29) proves less efficient than in IPA or NMP, and globally leads to smaller nanosheets ($<50$ nm), along with some very large ones ($>500$ nm). In all the experiments the nanosheet composition was ascertained by EDX microanalysis (Supplementary Fig. 30).

UV-Vis-NIR spectroscopy of the colloid prepared from the 100 mg·ml$^{-1}$ powder dispersion in NMP was performed after drop-casting and drying of the liquid sample on a glass slide (Fig. 3f and Supplementary Fig. 31). This allows the removal of the solvent whose C-H bond intense absorption hinders light transmission in a wide NIR range. The resulting spectrum shows a continuous decrease of the absorption as the wavelength increases from 300 to 2,750 nm, followed by a wide absorption band from 2,750 to 3,300 nm, reaching a maximum at $\sim$2,900 nm (0.43 eV). Together with the absorption onset in

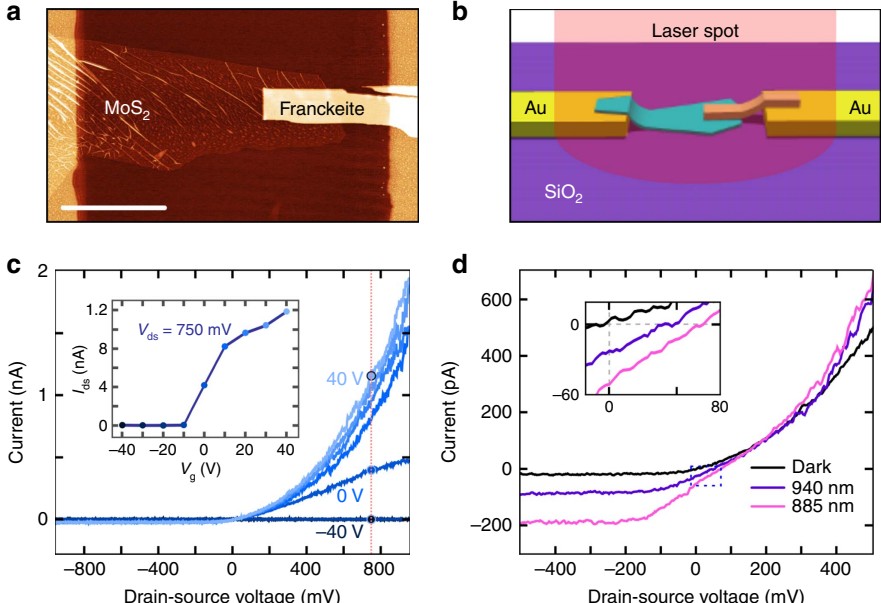

**Figure 5 | p-n junction made by stacking mechanically exfoliated flakes of MoS$_2$ and franckeite.** (**a**) AFM topographic image of the p-n junction. The scale bar is 10 μm. (**b**) Artistic representation of the p-n junction shown in **a**. (**c**) Diode-like current-voltage ($I_{ds} - V_{ds}$) curve of the p-n junction in dark conditions for different applied back-gate voltages. Inset: gate trace extracted from the $I_{ds} - V_{ds}$ at $V_{ds} = 750$ mV. The p-n junction switches on at an applied back-gate voltage of 0 V. (**d**) Diode-like current-voltage ($I_{ds} - V_{ds}$) curve of the p-n junction at an applied back-gate voltage of $V_g = 40$ V in dark conditions and upon illumination with lasers of 940 and 885 nm wavelengths, both with a power of 140 μW. The inset highlights the region around $V_{ds} = 0$ V and $I_{ds} = 0$ V to show the short-circuit current ($I_{sc}$) and open circuit voltage ($V_{oc}$) values, obtaining $I_{sc} = -27$ pA and $V_{oc} = 55$ mV at 940 nm, and $I_{sc} = -51$ pA and $V_{oc} = 77$ mV at 885 nm.

the NIR region, the existence of this band is consistent with the narrow bandgap energy determined for franckeite.

The Raman characterization was performed on bulk franckeite powder and on the most concentrated colloid obtained in NMP. Both samples present similar spectra, with five main bands centred at 66, 145, 194, 253, 318 cm$^{-1}$, and a shoulder from $\sim$400 to 650 cm$^{-1}$ (Fig. 3g), confirming the nature of the colloid obtained. As in a first approximation, franckeite alternates SnS$_2$ and PbS layers having a crystal structure similar to that of SnS$_2$ and PbS (see Fig. 1), most of its Raman bands can be associated to those of the parent structures (Supplementary Table 4 and Supplementary Note 9). However, the band at 253 cm$^{-1}$ would result from the combination of phonon modes of both the Q and H layers[30]. We also find differences in the relative intensities between the Raman spectra of bulk and exfoliated materials, probably originating from their respective thickness.

Taken together, these results demonstrate that franckeite undergo efficient LPE in NMP, IPA/water (1/0, 1/1, 1/4 mixtures) and other solvents resulting in few-layer nanosheets. The use of NMP leads to more stable and potentially more concentrated suspensions, but produces thicker nanosheets together with tiny nanoparticles. In contrast, LPE in IPA/water yields thinner nanosheets with larger area, but the suspensions are less stable.

**Franckeite-based nanodevices.** To further explore the electronic properties of franckeite, we have employed mechanically exfoliated flakes in the fabrication of electronic devices by transferring the flakes onto Ti/Au electrodes pre-patterned on a SiO$_2$/Si substrate. The flakes are placed bridging the electrodes using a deterministic transfer technique with an all dry viscoelastic material[31]. Figure 4a shows an AFM topographic image of one of these devices (with a thickness ranging from $\sim$7 nm to $\sim$13 nm, see Supplementary Figure 36 for optical microscopy characterization). The devices are characterized by measuring

current-voltage characteristics (the current passing through the material ($I_{ds}$) while sweeping the drain-source voltage ($V_{ds}$) with a fixed back-gate voltage ($V_g$)) and by measuring the current dependence on the back-gate voltage with a fixed drain-source voltage (transfer curve). Figure 4b shows the $I_{ds} - V_g$ curve for the device shown in Fig. 4a measured in dark conditions, high vacuum ($P < 10^{-5}$ mbar) and with an applied $V_{ds} = 150$ mV. The dependence of the current on the back-gate voltage serves as a test for determining the doping of the material: the decrease in the current with increasing back-gate voltage indicates that the material is p-doped in agreement with the STS and thermopower measurements for bulk. The gate traces also show that the franckeite flake is strongly doped and it cannot be switched off within the experimental gate voltage window. Thus, franckeite does not seem an appropriate material to fabricate FETs and requires doping engineering to reduce the intrinsic doping. This measurement was repeated after 41 days, finding a small drop of 5% in the current intensity, which indicates that the device remains stable over time. This is very relevant when comparing with black phosphorus, the other intrinsically doped p-type 2D semiconductor, which degrades on a time scale of a few hours[32]. From the $I_{ds} - V_{ds}$ curve (shown in the inset of Fig. 4b) we estimate a resistivity of $\sim$50 mΩ·m.

The absorption spectroscopy of the liquid-phase-exfoliated material and the STS measurements on bulk franckeite suggest that it presents a narrow bandgap and therefore motivates the application of franckeite as a photodetector working in the NIR (it should be able to generate photocurrents upon illumination with light wavelength as long as $\sim$3,000 nm). To test the optoelectronic characteristics of our franckeite-based photodetectors, we first study the dependence of the $I_{ds} - V_g$ curves on the illumination with a laser source. The measurements, plotted in Fig. 4c, are carried out using a laser of 640 nm wavelength and show that the drain-source current of the photodetector increases with increasing light power over the full range of the gate voltage.

In order to further characterize the photodetector we calculate the responsivity, a typical figure-of-merit for photodetectors that represents the input-output gain of the device as a function of the laser effective power reaching the device (Fig. 4d). The responsivity ($R$) is calculated as $R = I_{ph}/P_{eff}$, where $I_{ph}$ (photocurrent) is the difference between the current measured upon illumination and in dark conditions, and $P_{eff}$ is the effective power of the laser that reaches the device ($P_{eff} = P_{laser} \cdot A_{device}/A_{spot}$). For the device shown in Fig. 4a, upon illumination at 640 nm, we obtain a maximum responsivity of $\sim 100\ mA \cdot W^{-1}$ for a laser intensity of $\sim 30\ mW \cdot cm^{-2}$. Even if this value is not as high as for other two-dimensional photodetectors, such as monolayer $MoS_2$ ($R > 10^6\ mA \cdot W^{-1}$) or $In_2Se_3$ ($R > 10^7\ mA \cdot W^{-1}$) (refs 33–35), it is larger than most of the responsivities measured in few-layer black-phosphorus, the two-dimensional phototedector with the narrowest bandgap reported to date[36,37], ranging from $0.5\ mA \cdot W^{-1}$ to $135\ mA \cdot W^{-1}$ (refs 38–40).

We have also studied the photocurrent generation in franckeite photodetectors upon illumination in a wide range of light wavelengths (from 405 nm (UV) to 940 nm (NIR)). The $I_{ds} - V_g$ curves, measured in dark conditions and upon illumination with lasers of different wavelengths (Fig. 4e), reveal that the device is able to generate photocurrent at wavelengths as large as 940 nm, in good agreement with the results obtained from the UV-Vis-NIR spectroscopy of the liquid-phase-exfoliated material. The photocurrents calculated from these measurements for two fixed back-gate voltages are plotted in Fig. 4f. We address the reader to Supplementary Notes 10–13 and Supplementary Figs 32 and 36–40 for time response characterization of these devices, the full characterization of another device and the electronic characterization of a device based on liquid-phase-exfoliated franckeite.

**MoS$_2$-franckeite p-n junction**. As a proof-of-concept, we employ one of the most well-known n-type two-dimensional materials, $MoS_2$, in combination with franckeite to fabricate a p-n junction (building blocks of electronics) based on the stacking of mechanically exfoliated franckeite and $MoS_2$ flakes. Figure 5a shows an AFM topographic image of the device (see Supplementary Figure 41 for optical microscopy characterization and AFM height profile), which is represented in an artistic drawing in Fig. 5b: the $MoS_2$ flake is first deposited by deterministic transfer onto a $SiO_2$ substrate in contact with one pre-patterned Ti/Au electrode, then a franckeite flake is deposited in contact with the other pre-patterned electrode, resulting in a van der Waals heterostructure made of a p-doped material (franckeite) and an n-doped material ($MoS_2$) in the overlapping region. The electronic characterization of the device, carried out in vacuum ($P < 10^{-5}$ mbar), at room temperature and in dark conditions, shows diode-like $I_{ds} - V_{ds}$ characteristics for different back-gate voltages (Fig. 5c). The $I_{ds} - V_g$ curve shown in the inset of Fig. 5c yields a current rectification ratio of 400 and a gate threshold voltage of $V_{th} \sim -10$ V. To test the optoelectronic properties, the device is illuminated as represented in Fig. 5b with laser spots of 940 and 885 nm wavelengths, indicating that there is photocurrent generation even for zero applied voltage (short circuit current, $I_{sc}$) and that the current is zero for a finite positive applied voltage (open circuit voltage, $V_{oc}$). This phenomena is due to the photovoltaic effect: upon illumination at zero applied voltage, the photogenerated electron-hole pairs are separated by an internal electric field, generating a photocurrent ($I_{sc}$) with the same sign as the reverse voltage; on the other hand, charge carriers are accumulated at different parts of the device, creating a voltage when the circuit is open ($V_{oc}$) in the forward voltage direction. The photocurrent measured upon illumination with laser spots of 940 and 885 nm wavelengths (Fig. 5d) presents

$I_{sc} = -27$ pA and $V_{oc} = 55$ mV at 940 nm, and $I_{sc} = -51$ pA and $V_{oc} = 77$ mV at 885 nm (see Supplementary Note 14 and Supplementary Figs 41–45 for a more detailed analysis of the characteristics of the device). We should stress here that optimizing the performance of franckeite-based p-n junction devices is beyond the scope of this work. Nevertheless, these results demonstrate that one can exploit the p-type character of franckeite in electronic devices where a narrow gap air-stable p-type semiconductor is needed.

## Discussion

In summary, we have shown that bulk franckeite can be exfoliated both mechanically and in liquid phase to afford the first naturally occurring quasi 2D van der Waals heterostructure. The structure and properties of ultrathin flakes of franckeite have been studied extensively from both theory and experiment. Franckeite nanosheets show a very narrow bandgap $<0.7$ eV and p-type conductivity, and are highly stable under ambient conditions, both as mechanically exfoliated flakes and as colloidal suspensions. These features make it a unique addition to the still rather small library of experimentally investigated 2D materials. As validation for its potential technological application, we have constructed prototype photodetectors based on mechanically exfoliated few-layers crystals, as well as a p-n junction made by stacking an $MoS_2$ flake and a franckeite flake.

## Methods

**Materials.** Bulk franckeite mineral from mine San José, Oruro (Bolivia) was used for both mechanical and chemical exfoliation. All the experiments were carried out with flakes obtained from the same crystal.

**Density functional theory calculations.** Calculations of the electronic properties of franckeite are based on the framework of DFT, as implemented in the Quantum ESPRESSO package[21]. The GGA of Perdew-Burke-Ernzerhof (GGA-PBE) was adopted for exchange-correlation functional[41]. Part of the calculations were also performed using the hybrid nonlocal exchange-correlation treatment that incorporated 25% screened Hartree-Fock exchange, the HSE06 functional[42]. The HSE functional, with its fraction of screened short-ranged Hartree-Fock exchange, yields reasonably accurate predictions for energy band gaps in semiconductors[43,44]. The electron-ion interaction employed in the calculations is described using the norm-conserving Troullier-Martins pseudopotentials[45]. The energy cut-off for the plane wave basis set is set to 60 Ry with a charge density cut-off of 240 Ry. We have used a Monkhorst-Pack scheme with a $5 \times 5 \times 3$ k-mesh for the Brillouin zone integration for the supercell (22 atoms) (ref 46). Herein, we employed the van der Waals interaction described within a semiempirical approach following the Grimme formula[47]. In some cases, the spin-orbit coupling, important for the heavy elements considered here, is included in the self-consistent calculations of electronic structure.

**Electron microscopies and XPS of mechanically exfoliates flakes.**
*HRTEM, SAED and SEM.* Mechanically exfoliated franckeite layers were transferred onto a holey $Si_3N_4$ membrane window grid and characterized using a JEOL JEM 3000F microscope (TEM, 300 kV) and observed with a Zeiss EVO MA15 microscope (SEM, 15 kV).

*XPS.* LEEM and micro-XPS measurements were done at the LEEM microscope (Elmitec, GmbH) in operation at ALBA synchrotron (Barcelona, Spain)[24]. The instrument is equipped with a $LaB_6$ electron gun for real space imaging (LEEM), with lateral resolution of 20 nm. The imaging column of the microscope has an electron energy filter, and using the photons of tunable energy coming from the synchrotron (at 16° incidence angle), it is possible to perform XPS with an energy resolution of 0.25 eV keeping the lateral resolution.

**Liquid-phase exfoliation.** *Colloid preparation.* Fragments of cleaved natural franckeite were grinded in a porcelain mortar until making a finely grained and black powder. This powder was dispersed in NMP (5 ml) at various concentrations (0.1, 1, 10 and 100 mg·ml$^{-1}$) in 20 ml glass vials. Each dispersion was sonicated for 1 h in a Fisher Scientific FB 15051 ultrasonic bath (37 kHz, 280 W, ultrasonic peak max. 320 W, standard sine-wave modulation) thermostated at 20 °C. Resulting black suspensions were centrifuged (990*g*, 30 min, 25 °C, Beckman Coulter Allegra X-15R, FX6100 rotor, radius 9.8 cm) to remove poorly exfoliated solid. After centrifugation, the corresponding supernatants were carefully isolated from the black sediments to obtain exfoliated franckeite pale-to-dark orange colloids respectively, stable over at least 6 months.

Exfoliation of $1\,mg\cdot ml^{-1}$ samples was carried out in similar conditions in IPA/water mixtures (1/0, 4/1, 1/1, 1/4, 0/1; v/v), methanol or DMF. LPE in IPA/water resulted in pale orange suspensions that precipitated at room temperature within 2 days (1/4 mixture), 4 days (1/1 mixture) or one week (pure IPA). Note that the nanosheets can easily be redispersed by short-time sonication (<1 min). Colloidal stability of the suspensions prepared in methanol or DMF compares to that in NMP (>2 months).

Comments on the stability of the suspensions and the control of the reproducibility of the exfoliation experiments can be found in Supplementary Note 8.

**AFM.** Colloidal suspensions were drop-casted on freshly cleaved mica substrates and dried under vacuum. Images were acquired using a JPK NanoWizard II AFM working in tapping (NMP) or contact (IPA/water) mode at room temperature in air.

**TEM.** Colloidal suspensions were drop-casted onto 200 square mesh copper grids covered with a carbon film and observed using a JEOL JEM 2100 microscope (200 kV), equipped with an Energy Dispersive X-Ray (EDX) detector.

**UV-Vis-NIR spectroscopy.** As-prepared colloidal suspension was drop-casted onto a microscope glass slide and dried at 60 °C. The sequence was repeated until a pale orange thin film appeared whose absorption was characterized using a Cary 5000 spectrophotometer, Agilent Technologies (wavelength range: 175–3,300 nm). The measurement was replicated three times with different glass slides.

**Raman spectroscopy.** Powder (pressed) and liquid-phase-exfoliated franckeite (dried at room temperature) deposited on glass slides were analysed with a WITec alpha300 RA combined confocal Raman imaging/atomic force microscope (objective NA 0.95, ×100; laser excitation: 532 nm, 0.2 mW). Powder- and exfoliated-material spectra result from the average of 200 and 5 measurements over the respective samples.

**Franckeite-based nanodevices.** *Sample fabrication.* Franckeite has been mechanically exfoliated using Nitto tape (SPV 224) on franckeite chips and a polydimethylsiloxane (PDMS) stamp (Gelfilm from Gelpak) afterwards. The mechanical exfoliation usually yields thick crystals (thickness larger than 100 nm) with lateral dimensions larger than 50 μm, although some thinner crystals (thickness between 10 and 20 nm) can be obtained from the first exfoliation with PDMS, with lateral dimensions in the range of 10–50 μm. The exfoliation with PDMS can be repeated one or two times more in order to obtain thinner crystals, but it is usually not needed. Then, the PDMS stamp with the desired flake is brought into contact with the two Au electrodes on SiO₂ and peeled off carefully, leaving the flake on the substrate. In this way we obtain three-electrodes devices (details about deterministic transfer can be found in ref. 31).

**AFM.** AFM characterization of the devices was carried out using a Digital Instruments D3100 AFM operated in the amplitude modulation mode.

**Electronic and optoelectronic characterization.** The electronic and optoelectronic characterization have been performed in a Lakeshore Cryogenics tabletop probe station at room temperature and high vacuum ($<10^5$ mbar). The light excitation is provided by diode pump solid state lasers operated in continuous mode and guided with an optical fibre which yields a spot of 200 μm in diameter on the sample.

**Data availability.** All data generated or analysed during this study are included in this published article (and its Supplementary Information Files).

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

## Acknowledgements

A.C.-G. acknowledges financial support from the BBVA Foundation through the fellowship 'I Convocatoria de Ayudas Fundacion BBVA a Investigadores, Innovadores y Creadores Culturales' ('Semiconductores ultradelgados: hacia la optpelectronica flexible'), from the MINECO (Ramón y Cajal 2014 program, RYC-2014-01406), from the MICINN (MAT2014-58399-JIN) and from European Commission under the Graphene Flagship, contract CNECTICT-604391. E.M.P. acknowledges financial support from the European Research Council (MINT, ERC-StG-307609) and from the MINECO of Spain (CTQ2014-60541-P). E.G. gratefully acknowledges the AMAROUT II fellowship program for receiving a grant for transnational mobility (Marie Curie Action, FP7-PEOPLE-2011-COFUND (291803)). A.J.M.-M., G.R.-B. and N.A. acknowledge the support of the MICCINN/MINECO (Spain) through the programmes MAT2014-57915-R, BES-2012-057346 and FIS2011-23488 and Comunidad de Madrid (Spain) through the programme S2013/MIT-3007 (MAD2D). J.O.I. and H.S.J.v.d.Z. acknowledge the support of the Dutch organization for Fundamental Research on Matter (FOM) and by the Ministry of Education, Culture, and Science (OCW). M.A.N. acknowlededges the support of the MICCINN/MINECO (Spain) through the programmes MAT2013-49893-EXP and MAT2014-59315-R. Authors M.A.N., A.J.M.-M. and A.C.-G. acknowledge the support from ALBA Synchrotron for the experiments performed at Circe beamline (BL24-CIRCE) at ALBA Synchrotron with the collaboration of ALBA staff (proposal ID 2015091399). W.S.P. acknowledges CAPES Foundation, Ministry of Education of Brazil, under grant BEX 9476/13-0. W.S.P. and J.J.P. acknowledge MICCINN/MINECO (Spain) for financial support under grant FIS2013-47328-C02-1; the European Union structural funds and the Comunidad de Madrid MAD2D-CM programme under grant nos. P2013/MIT-3007 and P2013/MIT-2850; the Generalitat Valenciana under grant no. PROMETEO/2012/011. W.S.P. and J.J.P. also acknowledge the computer resources and assistance provided by the Centro de Computación Científica of the Universidad Autónoma de Madrid and the RES.

## Author contributions

A.J.M.-M. and J.O.I. performed the optoelectronic characterization of the photodetectors and the p-n junction, fabricated by A.C.-G. E.G. performed the SEM, TEM, SAED characterization of the material and the liquid-phase exfoliation with the corresponding characterization. W.S.P. performed the DFT calculations of the band structure of franckeite. M.A.N., L.A., M.F., A.J.M.-M. and A.C.-G. performed the micro-XPS measurements on mechanically exfoliated flakes at Circe beamline (BL24-CIRCE) at ALBA Synchrotron with the collaboration of ALBA staff, and M.A.N. on the bulk material. A.J.M.-M., C.E. and N.A. performed the STM characterization of the bulk material. C.E. and N.A. performed the thermopower measurements of the bulk material. A.J.M.-M., E.G. and A.C.-G. wrote the manuscript. J.J.P., E.M.P. and A.C.-G. have directed the research project. All authors discussed the data and interpretation, and contributed during the writing of the manuscript. All authors have given approval to the final version of the manuscript.

## Additional information

**Additional information:** Supporting Information is available in the online version of the paper.  In the Supporting Information we include density functional theory calculations of thickness dependent bandgap, density functional theory calculations of the band structure with SOC, density functional theory calculations of the band structure with HSE06, density functional theory calculations of Sb-doped franckeite, scanning tunnelling microscopy characterization, X-ray diffraction characterization, scanning electron microscopy characterization, XPS on franckeite mineral chips, liquid-phase exfoliation, powder and liquid-phase exfoliated franckeite Raman spectra interpretation, devices based on liquid-phase exfoliated franckeite, thermopower, optical microscopy characterization of a franckeite photodetector, time response characterization of the franckeite photodetector, characterization of another franckeite photodetector and $MoS_2$-franckeite p-n junction.

**Competing financial interests:** The authors declare no competing financial interests.

