## [Peer Review File · Nature Communications]

Reviewers' Comments:

Reviewer #1 (Remarks to the Author)

The authors present a study of ultrathin layers of franckeite by combining DFT calculations and experiments. They demonstrate that franckeite flakes can be obtained by means of mechanical exfoliation and liquid-phase exfoliation. Based on the DFT calculations of the single Q layer, the single H layer and the franckeite crystal together with STS experiment on the franckeite crystal, the authors claim that franckeite nanosheets are p-type semiconductors with a bandgap < 0.7 eV. Moreover, they demonstrate potential applications of the franckeite layers in devices by building photodetectors based on the mechanically exfoliated systems and a p-n junction made of a franckeite flake and a n-type MoS₂. The study is interesting. While I appreciate the technical quality of the manuscript, I'm not convinced for the publication of its current form in Nat. Comm. because:

- 1) The system contains heavy element Pb, in which spin-orbital coupling (SOC) is important. However, it was not included in their DFT calculations. Moreover, the bulk phase has a number of atoms about 20 atoms. So, it is possible to perform a HSE calculation to see how the standard DFT underestimates the band gap of franckeite and its thin films, e.g., the single Q layer and H layer.
- 2) Despite the thickness of the franckeite layers in franckeite-based nanodevices is pointed out, it is not clear to me how thick they are in the section of Transmission electron microscopy and micro-XPS of mechanically exfoliated flakes
- 3) The connection between the theoretical calculations and experiment is weak. The flakes in the devices have a thickness in between 4 layers and 8 layers, however, DFT calculations were only performed for single Q and H layers. A systematic study of the thickness-dependence of the electronic structure not only makes the calculations more relevant to the experiment but also helps make a safe statement regarding the band gap.
- 4) A proper discussion of the underlying physical mechanism of rectification, which may be different from that for conventional p-n junction diodes, is recommended.

Reviewer #2 (Remarks to the Author)

A. The authors have provided theoretical and experimental characterization of mechanically and liquid phase exfoliated Franckeite material. They have provided details of the crystal structure, numerical calculation of energy bands of the basic material and by considering the influence of substitutional Sb or Fe atoms. They have presented TEM micrographs and the composition by XPS of mechanically exfoliated flakes. Further they demonstrate fairly stable liquid phase exfoliation of Franckeite starting from powders to nanosheets in 2 solvents. Then, fabricated electrical devices using mechanically exfoliated flakes for applications as near-infrared photodetectors and p-n junction heterostructures based on the stacking of MoS₂ and Franckeite.

B. The basic TEM, SEM and XPS characterization have been previously reported in the references provided by the authors and is not novel, however demonstrates the comprehensiveness of their measurements. The existence and properties of this material is also not new and have been known from a fairly long time. Only the concept of liquid phase exfoliation is novel and demonstrates the possibility of long term stability when compared to black phosphorous. It must be noted that it is not fully correct to state this comparison with black phosphorous since it degrades in few hours, instead must be compared to other 2D transition metal dichalcogenides (TMDs) of similar band gap. Also, no information on stable phases or their existence is provided which is quite important. Further in the application testing the use of these liquid phase exfoliated materials is absent and

not considered instead mechanically exfoliated flakes are used. Only the use of MoS₂ in forming heterostructures with Franckeite is unclear, there is no reason stated for choice of MoS₂ and not any other TMD.

C. Overall measurements and data are well presented with few doubts. The authors have not performed X-ray diffraction (XRD) measurements and which would strengthen the compositional insights of Franckeite. Minor details of crystal structure are also missing. It is not clear why the authors only present ways to liquid phase exfoliate Franckeite and their microscopic characteristics with no application based on them. And instead uses mechanically exfoliated flakes for studying electrical properties.

D. The only complaint of water based solvents used for liquid phase exfoliation needs to be rectified by providing comparison with other solvents and XRD measurements, otherwise all statistics seems appropriate.

E. All the conclusions are valid and reliable.

F. May include liquid phase exfoliation with other water based solvents to demonstrate further stability since only 2 solvents have been used for this study as it stands now. XRD measurements to strengthen the compositional details. Title can be improved and be more specific based on synthesis methods and applications developed.

G. The reference list can be improved and currently cites other similar 2D materials. There are more direct references on Franckeite in the literature than those currently cited. For instance, Eggleston et al. (2002), *American Mineralogist*, 87, 1273-1278 and Prior et al. (1904), *Mineralogical Magazine*, 14, 21-27.

H. The text is appropriate, clear, concise and well explained.

Reviewer #3 (Remarks to the Author)

The manuscript of Molina-Mendoza et al. reports a study of frackeite, a hoeterostructure of PbS and SnS₂. The authors include band structure calculations using density functional theory (DFT), analytical characterization (Raman, transmission electron microscopy, XPS), photocurrent measurements, and a heterostructure with MoS₂.

Although, quite a bit of data is included in the manuscript the case for novelty is unclear. To begin with the authors state the heterostructure has a gap of 0.7 eV, but the DFT calculations show an indirect band gap that is much smaller than 0.7 eV. Maybe this explains why the flakes are doped and little gate modulation is observed in a transistor. The photoresponse measurements are analyzed in terms a standard metrics (responsivity, Voc, Isc) but there is not much learning gained from these measurements. Lastly, the franckeite -MoS₂ heterostructure may not be as simple as the authors imply because there are three junctions here: Au-franckeite, franckeite-MoS₂, and MoS-Au.

Reviewer #4 (Remarks to the Author)

Molina-Mendoza et al.

The authors demonstrate the isolation of 4-7 layer thick crystallites of franckeite, a naturally occurring van der Waals heterostructure consisting of stacked layers of SnS₂ and PbS 2D materials. The authors use liquid phase exfoliation of bulk franckeite in NMP or water/IPA, with the most successful but less stable method being water/IPA. Few-layer franckeite is shown to be air stable, and is envisaged as a replacement for black phosphorous monolayers in applications.

The authors calculate the band structure of franckeite by DFT, and compare to STS measurements presented in the Supplementary information, finding that substitutional Sb is crucial for calculations to approach the experimentally measured band gap and observed p-doping.

Transmission electron microscopy is used to confirm features of the crystal structure first observed by Makovicky et al. in bulk and cited here. Micro XPS, EDS, Raman spectrometry and UV-Vis-NIR are all used to provide additional supporting materials characterisation.

The authors proceed to fabricate nanodevices based on mechanically exfoliated franckeite, including FET-type IV, photocurrent and diode-type IV measurements. The observations confirm the predicted bandgaps, and a promising photocurrent response at long wavelengths is shown.

Specific points:

The authors do not choose to fabricate nanodevices based on liquid phase exfoliated flakes - perhaps they could comment on whether this is due to a limitation of the liquid exfoliation method, such as flake size or cleanliness?

Given that mechanical exfoliation has been employed in the fabrication of few-layer franckeite here, could the authors provide comments about the success of this method, the number of thin flakes produced, whether it is possible to produce single layers of the franckeite structure, and if not why not.

In addition, the authors should state the thickness of the exfoliated franckeite flake in Fig 5a based on their topographic AFM data, and provide a scale bar for the topographic height. This data is presented in Fig 4a for the FET-type device.

The MoS₂/franckeite diode device clearly shows non-idealities in the MoS₂ flake, including wrinkles and trapped contamination - perhaps the authors could comment on the potential effect of such upon their measurements, despite the optimisation of such devices not being the stated goal of the present study.

Comments about the simplicity of the deterministic transfer method (line 240) in producing vdW heterostructures stand at odds to comments made in the introductory text (line 48).

Line 278 - could the authors clarify 'rather small library of 2D materials' ? Perhaps 'experimentally investigated 2D materials' is implied here, as large computational libraries of 2D materials already exist e.g. doi:10.1021/acs.jpcc.5b02950

I would suggest adding the following reference - doi:10.1038/ncomms11894 at line 48, as relevant to the present discussion and recently published in this journal.

Overall, I believe that the extensive characterisation and novelty of the isolation of the first naturally occurring van der Waals heterostructure merits publication in Nature Communications, in particular through demonstration of both liquid phase and mechanical exfoliation routes, providing the above points are addressed.

Reviewer #1 (Remarks to the Author):

The authors present a study of ultrathin layers of franckeite by combining DFT calculations and experiments. They demonstrate that franckeite flakes can be obtained by means of mechanical exfoliation and liquid-phase exfoliation. Based on the DFT calculations of the single Q layer, the single H layer and the franckeite crystal together with STS experiment on the franckeite crystal, the authors claim that franckeite nanosheets are p-type semiconductors with a bandgap < 0.7 eV. Moreover, they demonstrate potential applications of the franckeite layers in devices by building photodetectors based on the mechanically exfoliated systems and a p-n junction made of a franckeite flake and a n-type MoS₂. The study is interesting. While I appreciate the technical quality of the manuscript, I'm not convinced for the publication of its current form in Nat. Comm. because:

We thank the Reviewer for the careful reading of our manuscript and for his/her suggested changes. Below we have addressed the Reviewer's concerns and suggestions, point by point, in blue.

1) The system contains heavy element Pb, in which spin-orbital coupling (SOC) is important. However, it was not included in their DFT calculations.

We agree with the Reviewer that, as Pb is present in the Q layers, relativistic effects are expected to be large and have an appreciable effect on the electronic properties. To account for such relativistic effects, we have repeated some of the calculations reported in the manuscript including spin-orbit coupling (SOC). For the H layers, as shown in Fig. S2a, the inclusion of SOC did not change the band structure significantly, and, consequently, the band gap. On the other hand, in the case of Q layers and franckeite crystal (see Fig. S2b and S2c), the band gap has slightly decreased when SOC is taken into account. To be precise, as shown in Table I in the Supplementary Information, the direct gap at X has decreased by 0.20 eV in the PbSnS structure (Q layer) and the band gap at X has decreased by 0.15 eV in the franckeite structure. These changes are noticeable, but do not modify qualitatively the overall picture and we have chosen to report these only in the Supplementary material.

We also include the Sb-doped franckeite band structure in Figure S5.

“Density functional theory calculations of the band structure with SOC

As a heavy element such as Pb is present in the Q layers, relativistic effects are expected to be large and have an appreciable effect on the electronics properties. To account for such relativistic effects, spin-orbit coupling (SOC) is here included in the calculations with the GGA functional. For the H layers, as shown in Fig. S2a, the inclusion of SOC did not change the band structure significantly, and, consequently, the band gap. On the other hand, in the case of Q layers and franckeite crystal (see Fig. S2b and S2c), the band gap

has slightly decreased when SOC is taken into account. To be precise, as shown in Table I, the direct gap at X has decreased by 0.20 eV in the PbSnS structure (Q layer) and the band gap at X has decreased by 0.15 eV in the franckeite structure, almost closing the gap.

TABLE I. Fundamental band gaps (in eV) of H layer, Q layer, franckeite crystal and franckeite with 50% Sn and 50% Sb in the Q layer and 100% Sb in the H layer.

	GGA-PBE	GGA-PBE + SOC	HSE06
SnS ₂	0.95	0.95	2.00
PbSnS	0.50	0.30	0.70
Franckeite	0.50	0.35	0.75
Franckeite (Q-Sb/Sn and H-100% Sb)	0.20	0.03	0.50

Figure S5 | a. Calculated band structure of franckeite with 50% Sn and 50% Sb in the Q layer and 100% Sb in the H layer without considering SOC. **b,** The same but considering SOC.”

Moreover, the bulk phase has a number of atoms about 20 atoms. So, it is possible to perform a HSE calculation to see how the standard DFT underestimates the band gap of franckeite and its thin films, e.g., the single Q layer and H layer.

Following the Reviewer's suggestion (which is certainly appropriate), we have also used the hybrid nonlocal exchange-correlation functional (HSE06) in an attempt to get more reliable

values of the band gap. The results (without SOC) are given in Table I in the Supplementary material. The improvement in all cases is evident. The increased bandgap more than compensates the decrease due to SOC, given an overall finite gap for the franckeite crystal with and without Sb doping. For the single H layer, we obtain an indirect gap semiconductor with a bandgap at X of ~ 1 eV, which increases to ~ 2 eV using HSE06. For the Q layer our calculation yields a semimetal with a gap at X of ~ 0.5 eV (GGA) (~ 0.7 eV with HSE06). For the whole structure the gap at X also increases by 0.25 eV. We have added short comments concerning these results in the main text.

Main text

“For the H layer (Fig. 1b), using the standard generalized gradient approximation (GGA) to the functional, we obtain an indirect gap semiconductor with a bandgap at X of ~ 1 eV, which increases to ~ 2 eV using HSE06. For the Q layer (Fig. 1c) our calculation yields a semimetal with a gap at X of ~ 0.5 eV (GGA) (~ 0.7 eV with HSE06).

We now compute the band structure of the combined Q and H layers, i.e., the franckeite crystal, with the same composition as that of the individual layers, obtaining two sets of bands separated by a small gap. The value of the gap at X is now of ~ 0.5 eV (GGA) (see Fig. 1d) (~ 0.75 eV using HSE06)”

Supporting Information

“Density functional theory calculations of band structure with HSE06

Since GGA-PBE functionals usually underestimate the band gaps, we use here the hybrid nonlocal exchange-correlation functional (HSE06) in an attempt to get more reliable values, possibly closer to the experimental ones. The results without SOC are given in Table S1. The improvement in all cases is evident. The increased band-gap more than compensates the decrease due to SOC, given an overall finite gap for the the franckeite crystal with and without Sb doping.

Figure S2 | Electronic band structure of a, SnS₂ (H layer). b, PbSnS (Q layer) and c, bulk franckeite from DFT/PBE with spin-orbit coupling.”

2) Despite the thickness of the franckeite layers in franckeite-based nanodevices is pointed out, it is not clear to me how thick they are in the section of Transmission electron microscopy and micro-XPS of mechanically exfoliated flakes

We thank the Reviewer for pointing out this issue. To that matter, we have performed topographic characterization with AFM of the samples used in TEM and micro-XPS, finding that the average thickness in the samples measured with TEM is ~ 100 nm, while the mechanically exfoliated flake shown in Figure 2 of the main text and measured with micro-XPS presents a thickness ranging between 13 and 125 nm. We have added the corresponding AFM images in the Supporting Information:

Figure S9. a to c, AFM topographic images of the samples used to measured TEM in mechanically exfoliated flakes. The thickness of the different flakes ranges from 100 nm to 200 nm.

Figure S12. a, LEEM image of the flake used to perform the micro-XPS measurements in the main text (the field of view is $50\ \mu\text{m}$ and the electron energy is $0.12\ \text{eV}$), the red square indicates the region of integration where the XPS spectra has been acquired. b and c, AFM topographic images of the flake used to perform the micro-XPS measurements in the main text.

3) The connection between the theoretical calculations and experiment is weak. The flakes in the devices have a thickness in between 4 layers and 8 layers, however, DFT calculations were only performed for single Q and H layers. A systematic study of the thickness-dependence of the electronic structure not only makes the calculations more relevant to the experiment but also helps make a safe statement regarding the band gap.

We thank the Reviewer for pointing out this issue. Figure S1 shows the evolution of the calculated band structure with the number of layers. It is found that the band structure topology and the bandgap does not significantly change with thickness from 1L to 3L, being essentially preserved up to the bulk calculation with an infinite number layers. Thus, unlike most studied 2D semiconductors, where the reduction of the thickness is usually

followed by a sizable increase of the bandgap due to quantum confinement in the out-of-plane direction, bulk Franckeite has a monolayer-like band structure.

“Density functional theory calculations of thickness dependent band structure

Figure S1 shows the evolution of the calculated band structure with the number of layers. It is found that the band structure topology and the bandgap was not significantly changed with a thickness from 1L to 3L, being essentially preserved up to the bulk calculation with an infinite number layers.

Figure S1 | The evolution of the band structure of Franckeite with increased thickness for a $n = 1$. b, $n = 2$. c, $n = 3$ and d, bulk.”

4) A proper discussion of the underlying physical mechanism of rectification, which may be different from that for conventional p-n junction diodes, is recommended.

We thank the Reviewer for pointing out this issue and give us the opportunity to improve the discussion about our fabricated p-n junction. To further understand the rectification in our device, we have first fit the current-voltage characteristics of the junction in dark and without back-gate voltage, to a Shockley diode model including resistance in series and in parallel, finding that the ideality factor of the p-n junction is $n = 1.1$ with a series resistance $R_s = 1.68 \text{ G}\Omega$ and parallel resistance $R_p \rightarrow \infty$. This fit parameters suggests that the device works closely to an ideal Shockley diode, however, the high value in the series resistance also suggest that we might be dealing with contact resistance and Schottky barriers formed in the semiconductor-metal interface.

We have also performed the same analysis in the current-voltage characteristics for different applied back voltages, finding that the ideality factor deviates from the ideal Shockley diode values and that the series resistance decreases. This is probably due to the fact that the MoS₂ flake, which is underneath the franckeite flake, is much more affected by the electric field than franckeite, both because of both the lower thickness of the MoS₂ and the proximity of the flake to the substrate. This assumption is also supported by the fact that only positive applied back-gate voltages seem to increase the current passing through the device, while negative back-gate voltages decrease the drain-source current to almost zero, meaning that the behavior of the junction is not ambipolar, but more similar to an n-type semiconductor, i.e., the MoS₂ governs the transport through the junction.

We have added figures and discussion about this matter in the Supporting Information, together with height profile of the flakes measured by AFM.

Finally, we have also improved the discussion about the photoresponse of the device.

“In Fig. S41 we show AFM topographic and optical microscopy images of the p-n junction characterized in Fig. 5 of the main text, showing that the MoS₂ flake presents a thickness of ~ 1.4 nm, while the franckeite flake has a thickness of ~ 25 nm.

Figure S41. **a** and **b**, AFM and optical microscopy image of the p-n junction (MoS₂ - franckeite) characterized in Figure 5 of the main text. **c**, AFM topographic image showing the height profile of both flakes, i.e., MoS₂ and franckeite.

The p-n junction current-voltage characteristics have been fitted to a Shockley diode model with resistance in series (R_S) and in parallel (R_P) given by: [A. Ortiz-Conde *et al.*, *Solid-State Electron* 2000, **44**(10): 1861-1864]

$$I_{ds} = I_S \left[\exp \left(\frac{V_{ds} - I_{ds} R_S}{n V_T} \right) - 1 \right] + \frac{V_{ds} - I_{ds} R_P}{R_P}$$

Equation S1

where I_{ds} is the drain-source current passing through the junction, I_S is the saturation current, V_{ds} is the drain-source voltage, n is the ideality factor (with typically a value between 1 and 2) and $V_T = k_B T / e$ is the thermal voltage (with k_B the Boltzmann constant, T is the temperature and e the elementary charge). An explicit solution to equation S1 can be written in terms of the Lambert W-function:

$$I_{ds} = \frac{n V_T}{R_S} W \left\{ \frac{I_S R_S R_P}{n V_T (R_S + R_P)} \exp \left(\frac{R_P (V_{ds} + I_S R_S)}{n V_T (R_S + R_P)} \right) \right\} + \frac{V_{ds} - I_S R_P}{R_S + R_P}$$

Equation S2

For the drain-source current in dark at $V_g = 0$ V (Fig. S42), we obtain the values $I_S = 13.3$ pA, $n = 1.1$, $R_S = 1.68$ G Ω and $R_P \rightarrow \infty$. As shown, the experimental data fit very well to the Shockley diode model with parasitic resistance, and the extracted ideality factor (close to 1) suggests that our p-n junction works closely to an ideal Shockley diode. However, we have to consider here that Schottky barriers can be formed at the semiconductor-metal interface (MoS₂/Au and franckeite/Au), which are known to affect both the rectifying characteristics of the diodes and the resistance, this could explain the high value obtained for the series resistance. [M.S. Choi *et al.*, *ACS Nano* 2014, **8**(9): 9332-9340; Y. Deng *et al.*, *ACS Nano* 2014, **8**(8): 8292-8299]

Figure S42. Current-voltage characteristics in dark and $V_g = 0$. The experimental data (blue) have been fit to a Shockley diode model including parasitic resistance (red), obtaining the fitting parameters listed in the box.

We also observe by fitting the current-voltage characteristics of the p-n junction with positive applied gate voltages (Fig. S43) that the ideality factor increases with increasing gate voltage (reaching a maximum value of 4.3), suggesting a modulation of the charge densities and band alignment between the two semiconductors. Nevertheless, the transfer curve shown in the inset of Fig. 5c of the main text, in which the drain-source current increases with increasing gate voltage, indicates that the gate voltage is affecting mainly the MoS₂ flake. This situation was expected since we have shown that franckeite presents a rather weak field-effect (Fig. 4

in the main text) and, therefore, the electric field created by the gate voltage affects much more the carrier density in the MoS₂ flake than in the franckeite flake. As a consequence, the extracted value of R_S decreases with increasing gate voltage.

Figure S43. Current-voltage characteristics in dark at increasing back-gate voltage (from 10 V to 40 V). The experimental data (from light blue to dark blue) have been fit to a Shockley diode model including parasitic resistance (from light red to dark red).

The photoresponse of the device is analyzed upon illumination with light wavelengths of 940 nm and 885 nm (Fig. 5d in the main text). Here, a photovoltaic behavior is observed due to the presence of a short-circuit current (I_{SC}) and an open-circuit voltage (V_{OC}), which increase with increasing photon energy due to a higher absorption of the franckeite flake. The electrical power harvested in the device, calculated as $P_{el} = |I_{ds}| \cdot V_{ds}$, is represented in Fig. S44 for illumination with both light wavelengths, obtaining maximum values of $P_{el,max} \sim 0.5$ pW for 940 nm and $P_{el,max} \sim 1.2$ pW for 885 nm. Although these values are rather low when compared to other p-n junctions fabricated from two-dimensional materials, they serve as a good starting point to the development of optimized p-n junctions using franckeite as p-type semiconductor.

Figure S44 p-n junction electrical power harvested in the device, calculated as $P_{el} = |I_{ds}| \cdot V_{ds}$ upon illumination with a laser spot of **a** 940 nm wavelength with a $P_{el,max} \sim 0.5$ pW and **b** 885 nm wavelength with a $P_{el,max} \sim 1.2$ pW.

The current-voltage characteristics upon illumination for different applied gate voltage are represented in Fig. S45. The value of I_{SC} increases as the gate voltage increases. For $V_g < 0$ V we do not observe photocurrent, while for $V_g > 0$ V the photocurrent increases with increasing V_g . This is probably due to both a modulation of the band alignment of the two materials affected by the gate voltage and the increase of the carrier concentration (electrons) mainly in the MoS₂ flake that might reduce the carrier lifetime and give rise to enhanced recombination of photogenerated carriers. [M. Furchi *et al.*, *Nano Letters* 2014, **14**(8): 4785-4791]

Figure S45. p-n junction current-voltage characteristics for different back-voltages ranging from -40 V to 40 V upon illumination with **(a)** 940 nm wavelength and **(b)** 885 nm wavelength.”

Reviewer #2 (Remarks to the Author):

A. The authors have provided theoretical and experimental characterization of mechanically and liquid phase exfoliated Franckeite material. They have provided details of the crystal structure, numerical calculation of energy bands of the basic material and by considering the influence of substitutional Sb or Fe atoms. They have presented TEM micrographs and the composition by XPS of mechanically exfoliated flakes. Further they demonstrate fairly stable liquid phase exfoliation of Franckeite starting from powders to nanosheets in 2 solvents. Then, fabricated electrical devices using mechanically exfoliated flakes for applications as near-infrared photodetectors and p-n junction heterostructures based on the stacking of MoS₂ and Franckeite.

We thank the Reviewer for the careful reading of our manuscript and for his/her suggested changes. Below we have addressed the Reviewer's concerns and suggestions, point by point, in blue.

B. The basic TEM, SEM and XPS characterization have been previously reported in the references provided by the authors and is not novel, however demonstrates the comprehensiveness of their measurements. The existence and properties of this material is also not new and have been known from a fairly long time.

We agree with the Reviewer that franckeite has been studied previously in its bulk form, although there are not so many reports on this material and they are focused on characterization of its crystal structure and composition. According to Web of Science (Thomson Reuters, consulted on November 1st 2016) there are only 38 manuscripts related to franckeite in the last 70 years, and only 23 of them include franckeite in the title. Also, graphite was known before the isolation of graphene, and so was molybdenite before few layer transition metal dichalcogenides, most of the two-dimensional materials were known in bulk before the isolation of few layers.

Our work on franckeite is the first demonstration that this material can be exfoliated both by mechanical and in liquid-phase exfoliation, opening the door to study the rather unexplored and vast family of sulfosalts as low-dimensional materials. Furthermore, the proof of mechanical exfoliation in franckeite sets a new strategy to exfoliate naturally occurring van der Waals heterostructures, which up to date have only been produced artificially, thus presenting some difficulties like lattice matching or trapped adsorbates between layers. Especially, the liquid-phase exfoliation of franckeite creates new routes towards the production of suspended platelets consisting of the alternate stacking of two-dimensional layers, i.e., this is the first time that a van der Waals heterostructure has been exfoliated in liquid-phase.

We have also provided a proof-of-concept by employing franckeite in functional, air-stable NIR photodetectors and in combination with other two-dimensional materials to form a p-n junction, giving evidence that franckeite might also be interesting for applications requiring narrow-bandgap materials.

Overall, the novelty of our manuscript relies on the first study of a naturally occurring van der Waals heterostructure in a low-dimensional configuration, the study of a new family of low-dimensional materials (sulfosalts) and sets an starting point to study and to work with these heterostructures.

Only the concept of liquid phase exfoliation is novel and demonstrates the possibility of long term stability when compared to black phosphorous. It must be noted that it is not fully correct to state this comparison with black phosphorous since it degrades in few hours, instead must be compared to other 2D transition metal dichalcogenides (TMDs) of similar band gap.

We apologize because we do not fully understand this Reviewer's comment. Several TMDs have been explored up to date, but none of them present bandgaps in the near-infrared region of the electromagnetic spectrum, i.e., below 1 eV. All known TMDs present bandgaps in the visible region of the electromagnetic spectrum (usually between 1 and 3 eV). As a reference, we cite A. Castellanos-Gomez, *J. Phys. Chem. Lett.*, **2015**, 6 (21), pp 4280–4291, where the bandgap of most of the two-dimensional materials are reviewed and it is stated that the only two-dimensional material with a bandgap below 1 eV is black phosphorus. Therefore, we feel that an honest comparison between franckeite and other two-dimensional materials, regarding their bandgap, is with black phosphorus.

Figure 2 in A. Castellanos-Gomez, *J. Phys. Chem. Lett.*, **2015**, 6 (21), pp 4280–4291:

Figure 2. Comparison of the band gap values for different 2D semiconductor materials. The band gap values for conventional semiconductors have been also included for comparison.

The horizontal bars spanning a range of band gap values indicate that the band gap can be tuned over that range by changing the number of layers, straining, or alloying. In conventional semiconductors, the bar indicates that the band gap can be continuously tuned by alloying the semiconductors (e.g., $\text{Si}_{1-x}\text{Ge}_x$ or $\text{In}_{1-x}\text{Ga}_x\text{As}$). The range of band gap values required for certain applications have been highlighted at the bottom part of the figure to illustrate the potential applications of the different semiconductors

Also, no information on stable phases or their existence is provided which is quite important.

According to the Reviewer's suggestion, we have performed XRD measurements in order to identify the phases present in the material. The obtained XRD spectra is in good agreement with previously reported results and it has been added to the Supporting Information (presented at the Reviewer's comment C).

Further in the application testing the use of these liquid phase exfoliated materials is absent and not considered instead mechanically exfoliated flakes are used.

The Reviewer raises an interesting point in this comment. The liquid phase exfoliation of several two-dimensional materials has already been studied, however, their further application in electronic and optoelectronic devices remains challenging due to the low performance given by the fact that the exfoliated platelets are randomly deposited on the substrate and there is a poor contact between them. The most common techniques to fabricate devices from liquid-phase exfoliated material are inkjet printing, drop casting and spray coating and filtering and fishing, but, as already mentioned, they usually result in really low performance devices (F. Withers *et al.*, *Nano Lett.*, **2014**, *14* (7), pp 3987–3992), therefore, the fabrication of functional devices from liquid-phase exfoliated materials is still a subject of research for the community (F. Bonaccorso *et al.*, *Adv. Mater.*, **2016**, *28* (29), 6136–6166).

The liquid-phase exfoliation of franckeite in our manuscript focuses on the demonstration of the possibility to employ this technique in the production of franckeite-based suspensions and not in the optimization of the process in order to develop devices, which falls out the scope of the present manuscript. However, we have employed the drop casting technique to fabricate simple devices based on liquid-phase exfoliated franckeite (now included in the Supporting Information), although the performance of these devices is low compared to those fabricated with mechanically exfoliated flakes. Further studies in the liquid-phase exfoliation and the optimization of devices is a highly interesting direction to follow in future works.

“Devices based on liquid-phase exfoliated franckeite

We have investigated functional devices based on liquid-phase exfoliated franckeite fabricated by drop casting a suspension of franckeite in NMP (1.25 ± 0.25 mg/mL) onto Au electrodes pre-patterned on a SiO_2 substrate and let it dry at room temperature. The device was then annealed at 120 °C in vacuum ($P < 10^{-1}$ bar) during 4 hours in order to remove completely the solvent and thus to improve the contact between the platelets. The device is

shown in Fig. S32a, where the LPE-franckeite is covering the whole area, even the Au electrodes. Atomic force microscopy topographic characterization (Fig. S32b) yields a thickness of the franckeite layer of ~ 60 nm.

The suspension of franckeite in NMP was prepared as explained in the main text, and the concentration in exfoliated material was determined by weighing the dried residue from a known volume of the most concentrated colloid. The nanosheets were precipitated with chloroform, the dispersion was centrifuged (18,600 g, 15 min) and the sediment was dried in vacuum until no variation in its weight could be measured.

The electronic characterization of the device in dark conditions (Fig. S32c) yields a sheet resistance of ~ 86 G Ω /sq, with a conductivity of $\sim 10^{-6}$ S \cdot m $^{-1}$, with a conductivity of $\sigma \sim 10^{-6}$ S \cdot m $^{-1}$, which is just one order of magnitude smaller than the one obtained in inkjet-printed-based MoS₂ devices ($\sigma = 8.9 \cdot 10^{-5}$ S \cdot m $^{-1}$) [F. Withers *et al.*, *Nano Lett.*, **2014**, *14* (7), pp 3987–3992].

Figure S32. a, Optical microscopy image of the device based on liquid-phase exfoliated franckeite fabricated by drop casting a suspension of franckeite in NMP on Au electrodes pre-patterned on a SiO₂ substrate. b, AFM topographic characterization of the franckeite layer of the device shown in a. The thickness of the layer ranges between 19 nm and 63 nm. c, Electronic characterization of the device in dark conditions.”

Only the use of MoS₂ in forming heterostructures with Franckeite is unclear, there is no reason stated for choice of MoS₂ and not any other TMD.

We agree with the Reviewer that other two-dimensional materials such as TMDs could be used in combination with franckeite to fabricate p-n junctions. We have chosen MoS₂

because is a well-studied two-dimensional material that can be easily exfoliated and presents good transport properties and intrinsic n-type doping. We have also used this material due to the experience that we possess in our group to exfoliate mono- or few-layer flakes and its availability in our laboratory. Nevertheless, the use of other two-dimensional materials is an interesting suggestion and can be studied in future works. Regarding this Reviewer's concern, we have added a sentence in the main text:

“As a proof-of-concept, we employ one the most well-known n-type two-dimensional materials, MoS₂, in combination with franckeite to fabricate a p-n junction (building blocks of electronics) based on [...]”

C. Overall measurements and data are well presented with few doubts. The authors have not performed X-ray diffraction (XRD) measurements and which would strengthen the compositional insights of Franckeite. Minor details of crystal structure are also missing.

According the Reviewer's suggestion, we have performed XRD measurements in powder franckeite, finding good agreement with previously reported works. We have added the XRD spectra and a table with the different phases in the Supporting Information.

“X-ray diffraction characterization

Franckeite powder was studied by means of X-ray diffraction in a X'pert PRO $\theta/2\theta$ diffractometer (Panalytical) at room temperature in the facilities of Servicio Interdepartamental de Investigación (SIIdI), Universidad Autónoma de Madrid. The measured spectra is shown in Fig. S7, together with a fit for each peak. The diffraction peaks are in good agreement with previously reported values (listed in Table SX), [A. Mottana *et al.*, *Powder Diffraction* 2013, 7(2): 112-114] resembling a P2/m space group. [A. Mottana *et al.*, *Powder Diffraction* 2013, 7(2): 112-114]

Figure S7. X-ray diffraction pattern measured in powder franckeite.

Table S2. Diffraction peaks of the XRD pattern.

2θ (°)	Counts	hkl Ref.[A. Mottana et al. , Powder Diffraction 2013, 7(2): 112-114]
10.2215	230	002
15.3530	215	003
20.5132	877	004
25.7355	2701	005
30.9960	1957	020
33.8352	46	-
36.3505	40.72	-
37.0024	66	-116/-213
38.4193	66	204
39.0037	40	116
40.1041	50	-
40.4206	67	025
41.7489	152	008
43.7545	90	-
47.3494	157	-
50.0648	71	-
58.6927	79	00.11
64.6646	129	00.12
70.6909	49	-
98.5968	45	00.17

It is not clear why the authors only present ways to liquid phase exfoliate Franckeite and their microscopic characteristics with no application based on them. And instead uses mechanically exfoliated flakes for studying electrical properties.

We have characterized a device based on liquid-phase exfoliated franckeite. The characterization is addressed in a previous comment of the Reviewer.

D. The only complaint of water based solvents used for liquid phase exfoliation needs to be rectified by providing comparison with other solvents and XRD measurements, otherwise all statistics seems appropriate.

As recommended by the Reviewer, we have significantly extended the set of liquid-phase exfoliation experiments. Six new solvents have been investigated, namely:

isopropanol/water 4/1 and 1/1, as well as the corresponding pure solvents, to cover the range of mixtures between pure isopropanol and pure water; and two other polar coordinating solvents: methanol and *N,N*-dimethylformamide, to further explore the scope of our liquid-phase exfoliation process. All of the different suspensions have been characterized by UV-Vis-NIR spectroscopy and TEM imaging/EDX analysis. Additionally, the colloidal stability of the different suspensions is discussed in detail, with a specific work on the isopropanol/water series to distinguish between the proper exfoliation abilities and the dispersing properties of each solvent. Note that the experiments have been replicated at least three times each to check reproducibility. We comment on this supplementary work in the main text of the manuscript to stress the major differences arising from the use of distinct exfoliation media; the complete series of experiments and analyses are to be found in the “Liquid-phase exfoliation” section of the Supporting Information (Figures S13 to S29 and associated comments). The systematic EDX analysis of each exfoliated sample, together with the XRD characterization of the initial franckeite powder, ensured the composition of the observed exfoliated material.

Main text

“[...] LPE was also investigated in various isopropanol (IPA)/water mixtures, as well as in other two polar solvents, methanol and *N,N*-dimethylformamide (DMF). In the following, we will focus on NMP and IPA/water 1/4 (v/v), the latter matching surface tension with SnS₂. [J. Shen *et al.*, *Nano Letters* 2015, **15**(8): 5449-5454] Details on LPE in the other solvents and their comparison are reported in the Supporting Information. [...]”

[...] LPE in methanol or DMF (Fig. S26-S29) proves less efficient than in IPA or NMP, and globally leads to smaller nanosheets (<50 nm), along with some very large ones (>500 nm). In all experiments, nanosheet composition was ascertained by EDX microanalysis (Fig. S30). [...]”

Supporting Information

“LPE in NMP

Figure S13 | Franckeite nanosheet colloidal suspensions in NMP. a, From left to right: colloidal suspensions obtained after sonication of 0.1, 1, 10 and 100 mg·mL⁻¹ franckeite powder dispersions in NMP and centrifugation. **b,** Extinction spectra of the colloidal suspensions shown in **a**.

LPE in IPA/water mixtures

Exfoliation experiments in various IPA/water mixtures (1/0, 4/1, 1/1, 1/4, 0/1) were carried out as described in the “Methods” section: 1 h bath sonication of $1 \text{ mg}\cdot\text{mL}^{-1}$ franckeite dispersions in the chosen mixture, followed by centrifugation to get a colloidal suspension of franckeite nanosheets in the proper mixture. Optical images and extinction spectra of the corresponding samples show that only IPA, the 1/1 and the 1/4 mixtures are able to both produce and disperse exfoliated nanosheets, the latter to a lesser extent (Fig. S17a, S18a). To better compare and precise the exfoliation efficiency within the series, the sediments originating from the sonication-centrifugation sequences of each sample were redispersed (1 min sonication) in IPA. The previous observations indeed proved that IPA is a good dispersing medium for franckeite nanosheets and particles. After centrifugation of the sediment dispersion in IPA, new colloidal suspensions are obtained that contain exfoliated material initially produced but not always properly dispersed in the original IPA/water mixtures. Figs. S14b, S18b-f and S19a show optical images and the corresponding extinction spectra of these various IPA redispersions. From these results, it appears that exfoliation is occurring in IPA/water 4/1 and water, although in the latter case scattering is dominating the extinction spectrum and the quantity of produced material is very low. Exfoliated material can also be recovered from the other samples, in similar proportions to that obtained from IPA/water 4/1. To be noticed is the particularly high amount of material recovered from the IPA/water 1/4 sediment; cumulated with the material initially dispersed in this mixture, the total amount of exfoliated material compares with that achievable in the best dispersing media (*i.e.* IPA and IPA/water 1/1; Fig. S18b, d, e and S19b). As almost no nanosheets were detectable in the 4/1 mixture and in water, the cumulated exfoliated franckeite in those cases amounts to the material recovered in IPA, which represents, in the 4/1 mixture case, the half of the material produced in IPA, IPA/water 1/1 or 1/4. To sum up, in our conditions: IPA and IPA/water 1/1 are both efficient media in terms of exfoliation of franckeite and dispersion of the resulting nanosheets; on the contrary, IPA/water 4/1 and water do not disperse franckeite nanosheets and produce relatively weak amount of exfoliated material; as a particular case, IPA/water 1/4, in spite of its substantial water content, is able to disperse noticeable quantities of nanosheets and produce a significant amount of exfoliated material. These remarkable properties most probably relate to IPA/water 1/4 specific surface tension that matches that of SnS_2 .

Figure S17 | Franckeite nanosheet colloidal suspensions from exfoliation in IPA/water mixtures. a, From left to right: nanosheet suspensions obtained after sonication of $1 \text{ mg}\cdot\text{mL}^{-1}$ franckeite powder dispersions in IPA/water 1/0, 4/1, 1/1, 1/4 and 0/1, and subsequent centrifugation. **b**, From left to right: nanosheet suspensions obtained after IPA

redispersion of the sediments from the experiments in IPA/water 1/0, 4/1, 1/1, 1/4 and 0/1, and subsequent centrifugation.

Figure S18 | UV-Vis-NIR spectroscopy of the colloidal suspensions prepared by sonication of $1 \text{ mg}\cdot\text{mL}^{-1}$ franckeite dispersions in various IPA/water mixtures (v/v). Extinction spectra of the suspensions obtained after sonication of franckeite powder in the mixture indicated (7 mL) and centrifugation (solid line); extinction spectra of the suspensions obtained after redispersion of the sediment resulting from the centrifugation in the same volume of IPA (dotted line); sum of the two preceding spectra (dashed line).

Figure S19 | Comparison of the exfoliation processes in various IPA/water mixtures. a, Extinction spectra of recovered exfoliated franckeite in IPA from the sediment of each sample. **b,** Extinction spectra of cumulated exfoliated material from each experiment.

The TEM images of the samples prepared in IPA/water mixtures (Fig. S20-S25) deposited as colloid in the proper exfoliating solvent or as redispersed material in IPA in the relevant cases, evidence thin nanosheets that are globally very similar both in shapes and lateral sizes, relatively uniform and centered around 200 nm. Only the exfoliated material produced in the pure solvent show some distinct features: the IPA colloid includes an additional and significant amount of few-nanometer round-shaped nanoparticles (Fig. S20), whereas the few nanosheets recovered from the exfoliation in water are the only ones to exhibit such a damaged surface, with lots of defects (Fig. S25). Of particular interest is the comparison of Figs. S23 and S24 which shows that the nanosheets dispersed in the proper IPA/water 1/4 mixture or recovered in IPA from the corresponding sediment have exactly the same structure.

Figure S20 | TEM images of franckeite nanosheets obtained in IPA from a $1 \text{ mg}\cdot\text{mL}^{-1}$ powder dispersion. Nanosheets coming from the first colloidal suspension in IPA resulting from the centrifugation of the sonicated sample.

Figure S21 | TEM images of franckeite nanosheets obtained in IPA/water 4/1 v/v from a $1 \text{ mg}\cdot\text{mL}^{-1}$ powder dispersion. Nanosheets recovered from the IPA redispersion of the sediment resulting from the centrifugation of the sonicated sample.

Figure S22 | TEM images of franckeite nanosheets obtained in IPA/water 1/1 v/v from a $1 \text{ mg}\cdot\text{mL}^{-1}$ powder dispersion. Nanosheets coming from the colloidal suspension in IPA/water 1/1 resulting from the centrifugation of the sonicated sample.

Figure S23 | TEM images of francite nanosheets obtained in IPA/water 1/4 v/v from a $1 \text{ mg} \cdot \text{mL}^{-1}$ powder dispersion. Nanosheets coming from the colloidal suspension in IPA/water 1/4 resulting from the centrifugation of the sonicated sample.

Figure S24 | TEM images of franckeite nanosheets obtained in IPA/water 1/4 v/v from a $1 \text{ mg}\cdot\text{mL}^{-1}$ powder dispersion. Nanosheets recovered from the IPA redispersion of the sediment resulting from the centrifugation of the sonicated sample.

Figure S25 | TEM images of franckeite nanosheets obtained in water from a $1 \text{ mg}\cdot\text{mL}^{-1}$ powder dispersion. Nanosheets recovered from the IPA redispersion of the sediment resulting from the centrifugation of the sonicated sample.

LPE in pure polar solvents

LPE was carried out in methanol and DMF as well, with initial franckeite powder dispersions of $1 \text{ mg}\cdot\text{mL}^{-1}$, and compared to the results obtained in pure IPA, water and NMP in the same conditions. Figs. S26 and S27 show the optical images and extinction spectra of the corresponding colloids. From their observation, we can conclude that methanol and DMF roughly produce as high as half the amount of exfoliated material obtained with pure IPA or NMP. Nevertheless, their TEM micrographs reveal nanostructures that are very different from the previous ones. The colloid prepared in methanol is composed of thin and mainly small flakes, ca. 50-100 nm in size, together with some nanosheets reaching up to 500 nm (Fig. S28). As for the DMF sample, it contains mostly a mixture of small (50-100 nm) to very small (<20 nm) objects that look like nanoparticles more than nanosheets. Due to DMF dispersing abilities, thicker and micrometric flakes are also to be found (Fig. S29).

Figure S26 | Franckeite nanosheet colloidal suspensions from exfoliation in pure polar solvents. From left to right: nanosheet suspensions obtained after sonication of $1 \text{ mg}\cdot\text{mL}^{-1}$ franckeite powder dispersions in IPA, water, methanol, DMF and NMP, and subsequent centrifugation.

Figure S27 | UV-Vis-NIR spectroscopy of the colloidal suspensions prepared by sonication of $1 \text{ mg}\cdot\text{mL}^{-1}$ franckeite dispersions in various solvents. Extinction spectra of the suspensions obtained after sonication of franckeite powder in various solvents. **a.** Comparison of all pure solvents tested. **b.** Isopropanol (IPA). **c.** Water (H_2O). **d.** Methanol (MeOH). **e.** *N,N*-dimethylformamide (DMF). **f.** *N*-methylpyrrolidone (NMP).

Figure S28 | TEM images of franckeite nanosheets obtained in methanol from a $1 \text{ mg}\cdot\text{mL}^{-1}$ powder dispersion. Nanosheets coming from the colloidal suspension in methanol resulting from the centrifugation of the sonicated sample.

Figure S29 | TEM images of franckeite nanosheets obtained in *N,N*-dimethylformamide (DMF) from a $1 \text{ mg} \cdot \text{mL}^{-1}$ powder dispersion. Nanosheets coming from the colloidal suspension in DMF resulting from the centrifugation of the sonicated sample.

Suspension colloidal stability and reproducibility of the experiments

Colloidal stability of the suspensions depends on many parameters, but mainly on the dispersing abilities of the corresponding solvent, the weight (related to size and thickness) of the particles, and the temperature. All of these parameters are involved in our exfoliation process and are to be considered to explain the results obtained and ensure reproducibility. All samples prepared in NMP show long-term colloidal stability due to the strong coordination abilities of this solvent. Such a property also ensures highly reproducible exfoliation experiments because it can compensate for small variations in the size of exfoliated nanosheets during bath sonication or in the temperature during centrifugation. On the contrary some experiments, in IPA/water 1/4 for example, were found more difficult to replicate at the beginning, sometimes leading to particle-free supernatants after centrifugation. Nevertheless, redispersion in the same volume of a better dispersing solvent (NMP or IPA), and subsequent centrifugation (990 g, 30 min) consistently afforded stable colloids showing the same extinction properties as successfully prepared samples. We attributed this initially poor reproducibility in IPA/water 1/4 to the combined effects of the weaker dispersing abilities of this solvent, the large size of the nanosheets (ca. 200 nm) that greatly favors sedimentation, and an insufficient control of the temperature during the centrifugation step. Once better controlled the temperature during centrifugation, we reached the desired reproducibility. Nonetheless, due to nanosheet size, and IPA and water being respectively less and far less coordinating than NMP, the suspensions of this series suffer from lesser colloidal stability.

In the case of methanol and DMF, the control of the centrifugation temperature ensured the reproducibility of the exfoliation process. Once precipitated the micrometric heavier nanosheets (within ~24 h), the samples showed long-term stability thanks to the coordination properties of these solvents and the small flake size.”

E. All the conclusions are valid and reliable.

We thank the Reviewer for his/her assessment of our work.

F. May include liquid phase exfoliation with other water based solvents to demonstrate further stability since only 2 solvents have been used for this study as it stands now. XRD measurements to strengthen the compositional details.

Please see our answer to remark D about further liquid-phase exfoliation experiments in other water-based solvents and XRD measurements.

Title can be improved and be more specific based on synthesis methods and applications developed.

We thank the Reviewer for his/her suggestion on the title of the manuscript but we think that the current title highlights the major novelty of our work: the fact that we are studying a van der Waals heterostructure that is not man-made. Putting more information into the title might dilute this message, therefore we would prefer keeping it as it is. Nonetheless, if both the Editor and the Referee have a strong opinion on this respect and believe that a change of the title is necessary to guarantee the acceptance of the manuscript we will be, of course, willing to change it.

G. The reference list can be improved and currently cites other similar 2D materials. There are more direct references on Franckeite in the literature than those currently cited. For instance, Eggleston et al. (2002), *American Mineralogist*, 87, 1273-1278 and Prior et al. (1904), *Mineralogical Magazine*, 14, 21-27.

We thank the Reviewer for these suggested references. These references have been added to the text.

H. The text is appropriate, clear, concise and well explained.

We thank the Reviewer for his/her assessment of our work.

Reviewer #3 (Remarks to the Author):

The manuscript of Molina-Mendoza et al. reports a study of frackeite, a heterostructure of PbS and SnS₂. The authors include band structure calculations using density functional theory (DFT), analytical characterization (Raman, transmission electron microscopy, XPS), photocurrent measurements, and a heterostructure with MoS₂.

We thank the Reviewer for the careful reading of our manuscript and for his/her suggested changes. Below we have addressed the Reviewer's concerns and suggestions, point by point, in blue.

Although, quite a bit of data is included in the manuscript the case for novelty is unclear.

We agree with the Reviewer that franckeite has been studied previously in its bulk form, although there are not so many reports on this material and they are focused on characterization of its crystal structure and composition. According to Web of Science (Thomson Reuters, consulted on November 1st 2016) there are only 38 manuscripts related to franckeite in the last 70 years, and only 23 of them include franckeite in the title.

Franckeite is a material which has not been studied in a low-dimensional configuration, neither by mechanical or liquid phase exfoliation, therefore, our results provide the first approach in the study of a member of the vast and rather unexplored family of sulfosalts as low-dimensional materials. Furthermore, our work on franckeite provides the first study of a low-dimensional naturally occurring van der Waals heterostructure thus solving the difficulties associated to the artificial fabrication of van der Waals heterostructures, i.e., the alignment of crystal structures between layers or the trapping of adsorbate between layers. Also, the liquid-phase exfoliation of franckeite is the first exfoliation of a van der Waals heterostructure, since up to now, van der Waals heterostructures made by liquid-phase exfoliated materials could only be achieved by exfoliating separately different materials and combining them afterwards, what is known to be challenging and not available for every material (see, for example, F. Withers *et al.*, *Nano Lett.*, **2014**, *14* (7), pp 3987–3992).

We do not only proof the possibility of exfoliating franckeite by mechanical cleavage and in liquid-phase, but we also provide a proof-of-concept by employing mechanically exfoliated franckeite (and liquid-phase exfoliated franckeite in this revised version) in functional devices for NIR photodetection.

In general terms, this manuscript provides the first study of a naturally occurring van der Waals heterostructure in a low-dimensional configuration, the study of a new family of low-dimensional materials (sulfosalts) and sets an starting point to study and to work with these heterostructures.

To begin with the authors state the heterostructure has a gap of 0.7 eV, but the DFT calculations show an indirect band gap that is much smaller than 0.7 eV. Maybe this explains why the flakes are doped and little gate modulation is observed in a transistor.

We agree with the reviewer that the connection between the calculations and the experimental results might not fully match, but the DFT calculations are based on a bit simplified models regarding the actual composition of the franckeite crystal structure and they do not fully capture the compositional complexity of the material. However, we provide these calculations together with the experimental results in order to obtain a deeper understanding of franckeite. The experimental results yield an electronic bandgap of 0.7 eV measured in exfoliated franckeite by scanning tunneling spectroscopy, and we have also measured optical absorption up to a light wavelength of 2900 nm (~ 0.42 eV), suggesting a narrow optical bandgap. Our DFT calculations yielded smaller values of the gap, but this is a well-known problem with standard GGA calculations. In the new version of the manuscript we have included calculations with the more reliable functional HSE06 which actually increases the gaps and brings them closer to the experimental values. Regarding the doping of franckeite, the DFT calculations suggest that the presence of Sb atoms in the crystal structure might be responsible for opening a bandgap in the material and provide a p-type doping. Although we have only considered a few possibilities for Sb doping, the results point in the right direction. In addition, the presence of Sb in the material has been ascertained by XPS and the p-type doping has been demonstrated both by STS and electronic transport measurements. Also, the little gate modulation is most probably due to the strong doping in the material, although the decreasing current with increasing back-gate voltage evidences the p-type nature of this doping.

The photoresponse measurements are analyzed in terms of standard metrics (responsivity, Voc, Isc) but there is not much learning gained from these measurements.

We apologize because we do not fully understand this Reviewer's comment. The electronic and optoelectronic characterization of the mechanically exfoliated franckeite-based devices provide information about its doping, stability and performance. For instance, the decreasing current with increasing gate voltage in the transfer curve indicates that the material is intrinsically p-doped, the fact that the device cannot be switched-off is most probably due to the strong doping provided in by the Sb atoms (as mentioned in the main text) and the small change in the transfer curve after 41 days demonstrates that the device remains stable for long periods of time.

Regarding the optoelectronic characterization, the photocurrent generation shown in Figure 4 of the main text accounts for a photoconductive effect in the material in a wide range of the electromagnetic spectrum (from the UV to the NIR), also evidencing the narrow bandgap of the material, and the responsivity is a figure-of-merit widely used to characterize the performance of a photodetector that allows to compare between different photodetectors (M. Buscema *et al.*, *Chem. Soc. Rev.*, 2015, 44, 3691-3718).

Furthermore, we have also included in this revised version (Supporting Information) the time response characterization of the device shown in Figure 4 of the main text, where the slow time response accounts for a strong photogating effect taking place in two-dimensional photodetectors (M. Buscema *et al.*, *Chem. Soc. Rev.*, 2015, 44, 3691-3718; M.M. Furchi *et al.*, *Nano Lett.*, 2014, 14 (11), pp 6165–6170).

“Time response of the franckeite photodetector

In Fig. S37 we show the photocurrent generated by the franckeite device characterized in the main text as a function of time upon illumination with light wavelength of 640 nm and a modulated intensity of 2.78 W/m^2 . The rise time is $\sim 10 \text{ s}$, while the fall time is $> 100 \text{ s}$. This slow time response could be attributed to a strong photogating effect acting in our devices, where localized states are introduced within the bandgap of the material by impurities. Photogenerated charges can get trapped in these states and reside there for long times, hampering the electron-hole recombination and thus slowing down the time response of the device. [M. Buscema *et al.*, *Chem. Soc. Rev.*, 2015, 44, 3691-3718; M.M. Furchi *et al.*, *Nano Lett.*, 2014, 14 (11), pp 6165–6170]

Figure S37. Photocurrent as a function of time of the device characterized in Figure 4 of the main text. The rise time is $\sim 10 \text{ s}$, while the fall time is $> 100 \text{ s}$.”

Lastly, the franckeite -MoS₂ heterostructure may not be as simple as the authors imply because there are three junctions here: Au-franckeite, franckeite-MoS₂, and MoS-Au.

We thank the Reviewer for pointing out this issue and give us the opportunity to improve the discussion about our fabricated p-n junction. To further understand the rectification in our device, we have first fit the current-voltage characteristics of the junction in dark and without back-gate voltage, to a Shockley diode model including resistance in series and in parallel, finding that the ideality factor of the p-n junction is $n = 1.1$ with a series resistance $R_S = 1.68 \text{ G}\Omega$ and parallel resistance $R_P \rightarrow \infty$. This fit parameters suggests that the device works closely to an ideal Shockley diode, however, the high value in the series resistance also suggest that we might be dealing with contact resistance and Schottky barriers formed in the semiconductor-metal interface, as the Reviewer suggests.

We have also performed the same analysis to the current-voltage characteristics for different applied back voltages, finding that the ideality factor deviates from the ideal Shockley diode values and that the series resistance decreases. This is probably due to the fact that the MoS₂ flake, which is underneath the franckeite flake, is much more affected by the electric field than franckeite, both because of the lower thickness of the MoS₂ and the proximity of the flake to the substrate. This assumption is also supported by the fact that only positive applied back-gate voltages seem to increase the current passing through the device, while negative back-gate voltages decrease the drain-source current to almost zero, meaning that the

behavior of the junction is not ambipolar, but more similar to an n-type semiconductor, i.e., the MoS₂ governs the transport through the junction.

We have added figures and discussion about this matter in the supporting information, together with height profile of the flakes measured by AFM.

Finally, we have also improved the discussion about the photoresponse of the device.

“In Fig. S41 we show AFM topographic and optical microscopy images of the p-n junction characterized in Fig. 5 of the main text, showing that the MoS₂ flake presents a thickness of ~ 1.4 nm, while the franckeite flake has a thickness of ~ 25 nm.

Figure S41. **a** and **b**, AFM and optical microscopy image of the p-n junction (MoS₂ - franckeite) characterized in Figure 5 of the main text. **c**, AFM topographic image showing the height profile of both flakes, i.e., MoS₂ and franckeite.

The p-n junction current-voltage characteristics have been fitted to a Shockley diode model with resistance in series (R_S) and in parallel (R_P) given by: [A. Ortiz-Conde *et al.*, *Solid-State Electron* 2000, **44**(10): 1861-1864]

$$I_{ds} = I_S \left[\exp \left(\frac{V_{ds} - I_{ds} R_S}{n V_T} \right) - 1 \right] + \frac{V_{ds} - I_{ds} R_P}{R_P}$$

Equation S1

where I_{ds} is the drain-source current passing through the junction, I_S is the saturation current, V_{ds} is the drain-source voltage, n is the ideality factor (with typically a value between 1 and 2) and $V_T = k_B T / e$ is the thermal voltage (with k_B the Boltzmann constant, T is the temperature and e the elementary charge). An explicit solution to equation S1 can be written in terms of the Lambert W-function:

$$I_{ds} = \frac{nV_T}{R_S} W \left\{ \frac{I_S R_S R_P}{nV_T (R_S + R_P)} \exp \left(\frac{R_P (V_{ds} + I_S R_S)}{nV_T (R_S + R_P)} \right) \right\} + \frac{V_{ds} - I_S R_P}{R_S + R_P}$$

Equation S2

For the drain-source current in dark at $V_g = 0$ V (Fig. S42), we obtain the values $I_S = 13.3$ pA, $n = 1.1$, $R_S = 1.68$ G Ω and $R_P \rightarrow \infty$. As shown, the experimental data fit very well to the Shockley diode model with parasitic resistance, and the extracted ideality factor (close to 1) suggests that our p-n junction works closely to an ideal Shockley diode. However, we have to consider here that Schottky barriers can be formed at the semiconductor-metal interface (MoS₂/Au and franckeite/Au), which are known to affect both the rectifying characteristics of the diodes and the resistance, this could explain the high value obtained for the series resistance. [M.S. Choi *et al.*, *ACS Nano* 2014, **8**(9): 9332-9340; Y. Deng *et al.*, *ACS Nano* 2014, **8**(8): 8292-8299]

Figure S42. Current-voltage characteristics in dark and $V_g = 0$. The experimental data (blue) have been fit to a Shockley diode model including parasitic resistance (red), obtaining the fitting parameters listed in the box.

We also observe by fitting the current-voltage characteristics of the p-n junction with positive applied gate voltages (Fig. S43) that the ideality factor increases with increasing gate voltage (reaching a maximum value of 4.3), suggesting a modulation of the charge densities and band alignment between the two semiconductors. Nevertheless, the transfer curve shown in the inset of Fig. 5c of the main text, in which the drain-source current increases with increasing gate voltage, indicates that the gate voltage is affecting mainly the MoS₂ flake. This situation was expected since we have shown that franckeite presents a rather weak field-effect (Fig. 4 in the main text) and, therefore, the electric field created by the gate voltage affects much more the carrier density in the MoS₂ flake than in the franckeite flake. As a consequence, the extracted value of R_S decreases with increasing gate voltage.

Figure S43. Current-voltage characteristics in dark at increasing back-gate voltage (from 10 V to 40 V). The experimental data (from light blue to dark blue) have been fit to a Shockley diode model including parasitic resistance (from light red to dark red).

The photoresponse of the device is analyzed upon illumination with light wavelengths of 940 nm and 885 nm (Fig. 5d in the main text). Here, a photovoltaic behavior is observed due to the presence of a short-circuit current (I_{sc}) and an open-circuit voltage (V_{oc}), which increase with increasing photon energy due to a higher absorption of the franckeite flake. The electrical power harvested in the device, calculated as $P_{el} = |I_{ds}| \cdot V_{ds}$, is represented in Fig. S44 for illumination with both light wavelengths, obtaining maximum values of $P_{el,max} \sim 0.5$ pW for 940 nm and $P_{el,max} \sim 1.2$ pW for 885 nm. Although these values are rather low when compared to other p-n junctions fabricated from two-dimensional materials, they serve as a good starting point to the development of optimized p-n junctions using franckeite as p-type semiconductor.

Figure S44 p-n junction electrical power harvested in the device, calculated as $P_{el} = |I_{ds}| \cdot V_{ds}$ upon illumination with a laser spot of **a** 940 nm wavelength with a $P_{el,max} \sim 0.5$ pW and **b** 885 nm wavelength with a $P_{el,max} \sim 1.2$ pW.

The current-voltage characteristics upon illumination for different applied gate voltage are represented in Fig. S45. The value of I_{SC} increases as the gate voltage increases. For $V_g < 0$ V we do not observe photocurrent, while for $V_g > 0$ V the photocurrent increases with increasing V_g . This is probably due to both a modulation of the band alignment of the two materials affected by the gate voltage and the increase of the carrier concentration (electrons) mainly in the MoS₂ flake that might reduce the carrier lifetime and give rise to enhanced recombination of photogenerated carriers. [M. Furchi *et al.*, *Nano Letters* 2014, **14**(8): 4785-4791]

Figure S45. p-n junction current-voltage characteristics for different back-voltages ranging from -40 V to 40 V upon illumination with **(a)** 940 nm wavelength and **(b)** 885 nm wavelength.”

Reviewer #4 (Remarks to the Author):

Molina-Mendoza et al.

The authors demonstrate the isolation of 4-7 layer thick crystallites of franckeite, a naturally occurring van der Waals heterostructure consisting of stacked layers of SnS₂ and PbS 2D materials. The authors use liquid phase exfoliation of bulk franckeite in NMP or water/IPA, with the most successful but less stable method being water/IPA. Few-layer franckeite is shown to be air stable, and is envisaged as a replacement for black phosphorous monolayers in applications.

The authors calculate the band structure of franckeite by DFT, and compare to STS measurements presented in the Supplementary information, finding that substitutional Sb is crucial for calculations to approach the experimentally measured band gap and observed p-doping.

Transmission electron microscopy is used to confirm features of the crystal structure first observed by Makovicky et al. in bulk and cited here. Micro XPS, EDS, Raman spectrometry and UV-Vis-NIR are all used to provide additional supporting materials characterisation.

The authors proceed to fabricate nanodevices based on mechanically exfoliated franckeite, including FET-type IV, photocurrent and diode-type IV measurements. The observations confirm the predicted bandgaps, and a promising photocurrent response at long wavelengths is shown.

We thank the Reviewer for the careful reading of our manuscript and for his/her suggested changes. Below we have addressed the Reviewer's concerns and suggestions, point by point, in blue.

Specific points:

The authors do not choose to fabricate nanodevices based on liquid phase exfoliated flakes - perhaps they could comment on whether this is due to a limitation of the liquid exfoliation method, such as flake size or cleanliness?

The Reviewer raises an interesting point in this comment. We have demonstrated the possibility of exfoliating franckeite in liquid-phased using NMP and IPA/water as solvents. We have also investigated in this revised version other solvents in order to obtain a higher understanding of the liquid-phase exfoliation in franckeite. However, the fabrication of devices seems to be difficult, most probably because the drop casting technique lacks on the control of the deposition of exfoliated flakes. For most two-dimensional materials exfoliated in liquid-phase, the fabrication of devices is challenging and several methods

have been developed to that purpose employing different materials, solvents and deposition techniques (see, for example, F. Withers *et al.*, *Nano Lett.*, **2014**, *14* (7), pp 3987–3992) but, still, the devices produced by these methods present a really low performance, meaning that the fabrication of functional devices from liquid-phase exfoliated materials is still challenging for the community (F. Bonaccorso *et al.*, *Adv. Mater.*, **2016**, *28* (29), 6136–6166).

Following the Reviewer's suggestion, we have fabricated devices with liquid-phase exfoliated franckeite by drop casting a suspension of franckeite in NMP on Au electrodes pre-patterned on a SiO₂ substrate. Although we have been able to measure conductivity in the devices, they seem to not be sensible to incident light, therefore, the fabrication and optimization of devices based on liquid-phase exfoliated franckeite looks like a promising research line for future works.

We have included the electronic characterization of a device based on liquid-phase exfoliated franckeite in the Supporting Information:

“Devices based on liquid-phase exfoliated franckeite

We have investigated functional devices based on liquid-phase exfoliated franckeite fabricated by drop casting a suspension of franckeite in NMP (1.25 ± 0.25 mg/mL) onto Au electrodes pre-patterned on a SiO₂ substrate and let it dry at room temperature. The device was then annealed at 120 °C in vacuum ($P < 10^{-1}$ bar) during 4 hours in order to remove completely the solvent and thus to improve the contact between the platelets. The device is shown in Figure S32a, where the LPE-franckeite is covering the whole area, even the Au electrodes. Atomic force microscopy topographic characterization (Figure S32b) yields a thickness of the franckeite layer of ~ 60 nm.

The suspension of franckeite in NMP was prepared as explained in the main text, and the concentration in exfoliated material was determined by weighing the dried residue from a known volume of the most concentrated colloid. The nanosheets were precipitated with chloroform, the dispersion was centrifuged (18,600 g, 15 min) and the sediment was dried in vacuum until no variation in its weight could be measured.

The electronic characterization of the device in dark conditions (Figure S32c) yields a sheet resistance of ~ 86 G Ω /sq with a conductivity of $\sigma \sim 10^{-6}$ S·m⁻¹, which is just one order of magnitude smaller than the one obtained in inkjet-printed-based MoS₂ devices ($\sigma = 8.9 \cdot 10^{-5}$ S·m⁻¹). [F. Withers *et al.*, *Nano Lett.*, **2014**, *14* (7), pp 3987–3992]

Figure S32. a, Optical microscopy image of the device based on liquid-phase exfoliated franckeite fabricated by drop casting a suspension of franckeite in NMP on Au electrodes pre-patterned on a SiO₂ substrate. b, AFM topographic characterization of the franckeite layer of the device shown in a. The thickness of the layer ranges between 19 nm and 63 nm. c, Electronic characterization of the device in dark conditions.”

Given that mechanical exfoliation has been employed in the fabrication of few-layer franckeite here, could the authors provide comments about the success of this method, the number of thin flakes produced, whether it is possible to produce single layers of the franckeite structure, and if not why not.

We thank the Reviewer for pointing out this issue and gives us the opportunity to discuss about the mechanical exfoliation of franckeite. Mechanical exfoliation of franckeite resembles that of any other layered crystal: it produces a random distribution of thick and thin flakes. In the revised version we tried to quantify more the dimensions and thickness of the flakes that we typically produce. Regarding the single layers, we haven’t found flakes thinner than 7 nm so far. Regarding this matter, we have added a sentence in the Methods section:

“Franckeite-based nanodevices:

Sample fabrication. Franckeite has been mechanically exfoliated using Nitto tape (SPV 224) on franckeite chips and a polydimethylsiloxane (PDMS) stamp (Gelfilm® from Gelpak) afterwards. The mechanical exfoliation usually yields thick crystals (thickness larger than 100 nm) with lateral dimensions larger than 50 μm, although some thinner

crystals (thickness between 7 and 20 nm) can be obtained from the first exfoliation with PDMS, with lateral dimensions in the range of 10 to 50 μm .[...]"

In addition, the authors should state the thickness of the exfoliated franckeite flake in Fig 5a based on their topographic AFM data, and provide a scale bar for the topographic height. This data is presented in Fig 4a for the FET-type device.

We thank the Reviewer for pointing out this issue. According to the Reviewer's suggestion, we have added the height profile of the p-n junction in an AFM topographic image in the Supporting Information.

Figure S41. **a** and **b**, AFM and optical microscopy image of the p-n junction (MoS₂ - franckeite) characterized in Figure 5 of the main text. **c**, AFM topographic image showing the height profile of both flakes, i.e., MoS₂ and franckeite.

The MoS₂/franckeite diode device clearly shows non-idealities in the MoS₂ flake, including wrinkles and trapped contamination - perhaps the authors could comment on the potential effect of such upon their measurements, despite the optimisation of such devices not being the stated goal of the present study.

We agree with the Reviewer that wrinkles and adsorbates might hamper the performance of the device in our p-n junction. These defects usually yield a non-homogenous contact

between the two semiconductors in the p-n region, i.e., the contact will be “better or worst” depending on the presence of these defects, which is a common problem in artificially built van der Waals heterostructures that is difficult to overcome. Nevertheless, our studies about the photoresponse of the p-n junction are carried out with homogenous illumination in the whole device and, therefore, it does not account for local effects in the junction originating on these defects. A more systematic study, like scanning photocurrent in the junction, might lead to a better understanding of the effects of wrinkles and adsorbates, but the purpose of our p-n junction, although its performance is low, is a proof-of-concept to demonstrate that franckeite can be used as p-type semiconductor in combination with another two-dimensional material to obtain p-n junctions presenting photovoltaic effect, and a deep study of the different phenomena associated to the fabrication and performance of artificially built van der Waals heterostructures, as well as the optimization of the device, falls out of the scope of this manuscript.

Comments about the simplicity of the deterministic transfer method (line 240) in producing vdW heterostructures stand at odds to comments made in the introductory text (line 48).

According to the Reviewer’s suggestion, we have removed the mentioned sentence from the text.

Line 278 - could the authors clarify 'rather small library of 2D materials' ? Perhaps 'experimentally investigated 2D materials' is implied here, as large computational libraries of 2D materials already exist e.g. doi:10.1021/acs.jpcc.5b02950

We thank the Reviewer for this suggestion and we have accordingly changed this sentence.

I would suggest adding the following reference - doi:10.1038/ncomms11894 at line 48, as relevant to the present discussion and recently published in this journal.

We thank the Reviewer for this suggested reference about which we were not aware of. This reference has been added to the text.

Overall, I believe that the extensive characterisation and novelty of the isolation of the first naturally occurring van der Waals heterostructure merits publication in Nature Communications, in particular through demonstration of both liquid phase and mechanical exfoliation routes, providing the above points are addressed.

We thank the Reviewer for his/her assesment of our work.

Reviewers' Comments:

Reviewer #1 (Remarks to the Author)

Reviewer #2 (Remarks to the Author)

The claims presented by the authors in response to the questions are valid and fully justified. In particular, the data on XRD measurements and extension of liquid phase exfoliation provides a comprehensive study and relevant references have been made. The results seems appropriately novel for this material and explained with valid arguments. They are of interest to the 2D materials research community for a better understanding. Also, the discussions and reference relating to each point are added to the main and supplementary text.

The methodology/ recipe to exfoliate and analyze are valid too with good choice of solvent ratio for comparison. Regarding devices made by liquid phase exfoliated solution/flakes, dip coating would have been another optimal way to synthesize films, however other methods are valid too. Regarding the title, some tweaking may be performed for it to be more comprehensive. The work now is convincing for the publication in Nature Communications without further modifications. It would definitely influence existing knowledge of 2D materials by incrementing another unique compound. The statistical analysis is appropriate and provides good comparison with plausible reproducibility demonstrated.

Reviewer #4 (Remarks to the Author)

I thank the authors for their careful consideration of my comments and those of the other reviewers, which I read with interest, and also for addressing completely all of the point raised. I believe the manuscript is now very strong, and have no hesitation in recommending it for publication.

Reviewers' comments:

Reviewer #1 (Remarks to the Author):

I agree with Reviewer 2 that the methods used in this work are not novel but the system is new and interesting. The authors have satisfactorily addressed the issues I raised in my previous report. Substantial explanations have been added in the revised manuscript, which makes their arguments and conclusions convincing. In conclusion, I suggest publication of the manuscript.

We thank the Reviewer for his/her assessment of our work.

Reviewer #2 (Remarks to the Author):

The claims presented by the authors in response to the questions are valid and fully justified. In particular, the data on XRD measurements and extension of liquid phase exfoliation provides a comprehensive study and relevant references have been made. The results seems appropriately novel for this material and explained with valid arguments. They are of interest to the 2D materials research community for a better understanding. Also, the discussions and reference relating to each point are added to the main and supplementary text.

We thank the Reviewer for the careful reading of our manuscript and for his/her suggested changes. Below we have addressed the Reviewer's concerns and suggestions, point by point, in blue.

The methodology/ recipe to exfoliate and analyze are valid too with good choice of solvent ratio for comparison. Regarding devices made by liquid phase exfoliated solution/flakes, dip coating would have been another optimal way to synthesize films, however other methods are valid too. Regarding the title, some tweaking may be performed for it to be more comprehensive.

We have modified the title of our manuscript to fulfill the Nature Communications stile (no punctuation in the title), the new title is "Franckeite as a naturally occurring van der Waals heterostructure". We have decided not to change substantially the title as 3 out of 4 Referees found the original title appropriate.

The work now is convincing for the publication in Nature Communications without further modifications. It would definitely influence existing knowledge of 2D materials by incrementing another unique compound. The statistical analysis is appropriate and provides good comparison with plausible reproducibility demonstrated.

We thank the Reviewer for his/her assessment of our work.

Reviewer #4 (Remarks to the Author):

I thank the authors for their careful consideration of my comments and those of the other reviewers, which I read with interest, and also for addressing completely all of the point raised. I believe the manuscript is now very strong, and have no hesitation in recommending it for publication.

We thank the Reviewer for his/her assessment of our work.